# Variable and interactive effects of Sex, APOE ε4 and TREM2 on the deposition of tau in entorhinal and neocortical regions

Joseph Giorgio [1,2,10] ✉, Caroline Jonson[3,4,5,6,10], Yilin Wang[7,10], Jennifer S. Yokoyama [6,8], Jingshen Wang[9,11] & William J. Jagust [1,11] on behalf of the Alzheimer's Disease Neuroimaging Initiative*

The canonical Alzheimer's Disease (AD) pathological cascade posits that the accumulation of amyloid beta (Aβ) is the initiating event, accelerating the accumulation of tau in the entorhinal cortex (EC), which subsequently spreads into the neocortex. Here in a multi-cohort study (ADNI, A4, HABS-HD) of 1354 participants with multimodal imaging and genetic information we queried how genetic variation affects these stages of the AD cascade. We observed that females and APOE-ε4 homozygotes are more susceptible to the effects of Aβ on the primary accumulation of tau, with greater EC tau for a given level of Aβ. Furthermore, we observed for individuals who have rare risk variants in TREM2 and/or APOE-ε4 homozygotes there was a greater spread of primary tau from the EC into the neocortex. These findings offer insights into the function of sex, APOE and microglia in AD progression and have implications for determining personalised treatment with drugs targeting Aβ and tau.

Alzheimer's Disease (AD) follows a canonical cascade, with the accumulation of amyloid-beta (Aβ) being the primary event accelerating tau accumulation and spread from the entorhinal cortex (EC) into the neocortex leading to cognitive decline[1]. Substantial evidence supports the role of pathological Aβ aggregates as a primary event in both sporadic and genetically dominant forms of AD[2]. In particular, individuals with dominantly inherited AD mutations show a predictable and sequential evolution of AD pathophysiology with Aβ accumulation leading to tau accumulation and subsequent spread[3]. However, these genetically dominant forms of AD are rare and the evolution of sporadic late-onset AD is influenced by a multitude of genetic and lifestyle factors, with the relative penetrance of the canonical AD cascade profoundly affected by genetic variation and comorbidities[4,5].

Studying genetic variations that implicate discrete biological processes provides key insight into how physiological processes such as lipid processing, microglial activation, and sex influence the evolution of AD pathology. Further, to fully appreciate the complex and interacting processes that are involved in the evolution of AD pathology, genetic variations must be modelled together and in combination with AD biomarkers (i.e. Aβ and tau PET).

The ε4 allele of Apolipoprotein E (APOE) gene is the strongest risk factor for clinical sporadic late-onset AD, conferring an increased risk (i.e. odds ratio [OR]) of AD compared to APOE-ε3 homozygotes of up to 3.46 OR for ε4 heterozygotes and 13.04 OR for APOE-ε4 homozygotes[6], with APOE-ε4 homozygosity having a near complete penetrance for Aβ positivity[7]. Interestingly, the APOE-ε4 allele also has a marked increase

[1]Department of Neuroscience, University of California Berkeley, Berkeley, CA 94720, USA. [2]School of Psychological Sciences, College of Engineering, Science and the Environment, University of Newcastle, Newcastle, NSW 2308, Australia. [3]Center for Alzheimer's and Related Dementias, National Institutes of Health, Bethesda, MD 20892, USA. [4]DataTecnica LLC, Washington, DC 20037, USA. [5]Pharmaceutical Sciences and Pharmacogenomics Graduate Program, University of California, San Francisco, San Francisco, CA 94158, USA. [6]Memory and Aging Center, Department of Neurology, Weill Institute for Neurosciences, University of California, San Francisco, San Francisco, CA 94158, USA. [7]Department of Statistics and Actuarial Science, The University of Iowa, Iowa City, IA, USA. [8]Department of Radiology and Biomedical Imaging, University of California, San Francisco, CA, USA. [9]Division of Biostatistics, University of California Berkeley, Berkeley, CA 94720, USA. [10]These authors contributed equally: Joseph Giorgio, Caroline Jonson, Yilin Wang. [11]These authors jointly supervised this work: Jingshen Wang, William Jagust. *A list of authors and their affiliations appears at the end of the paper. ✉e-mail: jgiorgio@berkeley.edu

in the risk of developing AD in Aβ positive individuals, suggesting additional effects beyond Aβ[8]. Multiple lines of evidence link the APOE4 isoform to reduced homoeostatic clearing of Aβ[4,9–11]. This aberrant function of APOE4 occurs through dysfunction in multiple cell types in the brain that play diverse roles in AD pathogenesis, including neurons, microglia, and astrocytes[9,12]. APOE4 increases the accumulation of cortical Aβ by reducing the dissolution of soluble Aβ[13] and impairs the clearance of Aβ by disrupting the blood brain barrier[12]. Further, APOE4 has been implicated in increased tau accumulation, with increased levels of APOE4 resulting in increased tau phosphorylation and interneuronal spread of tau[9,12,14,15]. Therefore, APOE is not only involved in amyloidosis but also in downstream and parallel events such as the development of tau pathology.

Alongside the APOE-ε4 allele, rare genetic polymorphisms in Triggering Receptor Expressed on Myeloid Cells 2 (TREM2) have also been shown to be significant risk factors for sporadic late-onset AD[16]. TREM2 is a transmembrane protein expressed in microglia that performs critical functions in the immune response to AD pathology. TREM2 is involved in signalling cascades as well as the transition of microglia to a disease activated state, with a lack of functional TREM2 profoundly impacting microglia function[17–19]. Multiple TREM2 polymorphisms have been linked to an increased risk of AD with varying effect ranging from 1.2 to 3 OR, of which R47H with a 2.71 OR for clinical AD is the most widely studied[16,20]. Previous work investigating this variant in animal models shows that rare polymorphisms lead to a hypofunctional form of TREM2 promoting tau seeding and spreading[21]. Multiple other genetic variants on the TREM2 gene have been linked to increased risk of AD and negatively affect the function of TREM2 in vitro[16,18,19,22,23]. These variants may have an additive loss of function or alternatively may be in linkage disequilibrium with the same functional variant[24]. How trait variation in TREM2 impacts different phases of the AD pathological cascade is yet to be fully elucidated in humans and offers an approach to study how dysfunctional TREM2 impacts microglial functions leading to increased burden of Aβ and tau. Furthermore, it remains to be seen how TREM2 and APOE4 interact with each other and Aβ or tau leading to a greater burden of AD pathology.

Sex plays an important role in the pathogenesis of AD, with dementia incidence higher in females in late life[25]. Furthermore, females have consistently been shown to have higher tau tangle load at autopsy then men[26–28]. This post-mortem work is well supported in vivo with multimodal neuroimaging studies showing females are susceptible to higher levels and faster accumulation rates of tau for a given level of Aβ than their male counterparts[29–32]. Whether these increases are specific to Aβ influences on primary accumulation of tau in the EC, or tau spreading mechanisms from the EC into the neocortex is not well resolved. Furthermore, it is not clear if this finding in females is due to, or, exacerbated by varied immune responses such as TREM2-related microglial dysfunction, with interactions between sex and immune processes previously reported[33]. Finally, it is not well resolved how sex and the APOE-ε4 allele interact to increase AD risk with effects of age and dosage impacting this relationship[6], coupled with inconsistent in vivo imaging findings[30,32] further investigation is warranted.

Beyond a more complete appreciation of the biology governing the accumulation of AD pathology, understanding how genetic variation relates to variable development of AD pathologies is of pressing clinical need. The recent successful trials of Lecanemab and Donanemab have shown that reducing cortical amyloid-beta plaques (Aβ) led to significant slowing of cognitive decline; however, our grasp on the precise scope and conditions under which anti-amyloid immunotherapy delivers benefits remains unclear[34,35]. This is particularly pertinent as both Lecanemeb and Donenamab had varying treatment effects based on the primary outcome in females, and, both Lecanemab and Donanemab showing no significant effects of treatment on the primary outcome in homozygotes for the APOE-ε4 allele[34,35]. Therefore, it is critical to quantify the impact genetic variation has on the cascading effects of Aβ on medial temporal tau accumulation and subsequent spread into the neocortex to best understand who to treat with these drugs.

Here, we use causal path modelling to assess how genetic variation impacts the AD pathological cascade (Fig. 1). Using data from within subject multimodal PET and whole genome sequencing (WGS) in a sample of 1354 individuals we probe different stages of the AD cascade to understand how genetic variation in sex, APOE-ε4 and TREM2 exacerbate AD pathology. We tested the effect of genetic variation through pathways mediated by Aβ or via non-Aβ pathways, that is, primary tau accumulation in the EC and spread into the neocortex after accounting for Aβ. Our causal path is structured assuming that there are stages in the AD cascade, with the initial event being Aβ deposition, followed by Aβ-related deposition of tau in the EC, followed by tau spreading from the EC into the neocortex (Fig. 1). We hypothesise that each genetic variable affects the AD cascade through varied routes, interacting with either Aβ to increase levels of EC tau or with EC tau to increase levels of neocortical tau. Specifically, we hypothesise that sex will interact with Aβ to increase levels of EC tau and that the number of APOE-ε4 alleles will have an Aβ independent effect on tau. Through more exploratory analyses, we hypothesise

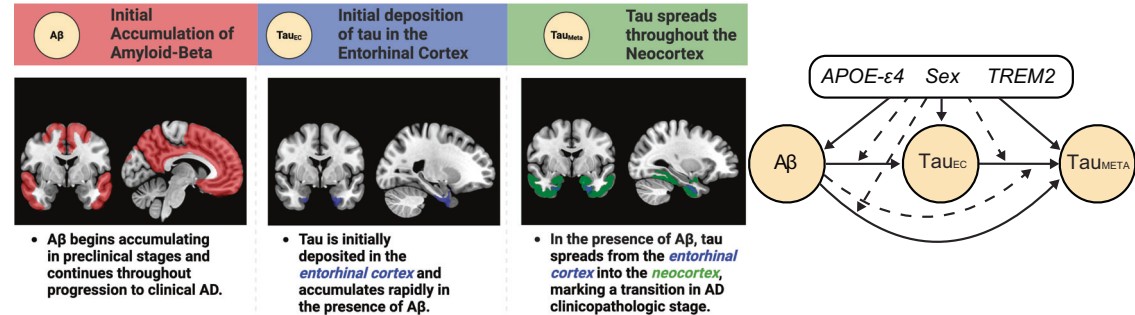

**Fig. 1 | Canonical Alzheimer's disease cascade.** The left panel indicates the stages and spatial distribution of Alzheimer's Disease (AD) pathology throughout the cascade. We modelled the initial stages of tau pathology using the entorhinal cortex (Tau_EC) region of interest. Neocortical tau was modelled using the meta-temporal (Tau_Meta) region of interest. The right panel shows the directed acyclic graph used to model the potential pathways between genetic variables and pathology. Each line originates at a predictor variable with the yellow nodes indicating an AD pathology. Solid lines indicate the pathway from an upstream variable to a downstream pathology, dashed lines indicate an interactive effect between upstream variables on a downstream pathology, with the arrow indicating a modulation of the pathway shown by the solid line. Solid and dashed lines initiating from the black genetic variables box indicate a direct or interactive effect, respectively, of genetic variable on the outcome variable. The black box indicates the three genetic variables that are modelled as direct or interactive predictors of AD pathologies, these are the number of Apolipoprotein E (APOE)-ε4 alleles (0,1,2), Sex (female, male), and Triggering Receptor Expressed on Myeloid Cells 2 (TREM2) risk variant carrier status (0,1).

**Table 1 | Sample demographics**

| Sample | Discovery | | | Replication |
|---|---|---|---|---|
| Cohort | ADNI | A4 | Combined | HABS-HD |
| n | 297 | 331 | 628 | 726 |
| Sex n(Female%) | 139(47%) | 199(60%) | 338(54%) | 433(60%) |
| Age mean ± std | 77.9 ± 7.2 | 71.7 ± 4.8 | 74.6 ± 6.8 | 65 ± 8.1 |
| Diagnosis (CN/MCI/AD) | 162/99/33[a] | 331/0/0 | 493/99/33[a] | 551/130/45 |
| Race (White/Hispanic/Black/Other) | 254/13/11/8[b] | 314/0/8/5[c] | 568/13/19/13[d] | 273/216/237/0 |
| Aβ CL mean ± std | 33.4 ± 43.2 | 51.9 ± 35.4 | 43.1 ± 40.3 | 13.4 ± 27.6 |
| EC Tau SUVR mean ± std[e] | 1.19 ± 0.26 | 1.18 ± 0.16 | 1.18 ± 0.21 | 1.32 ± 0.23 |
| MetaTemp Tau SUVR mean ± std^ | 1.23 ± 0.28 | 1.21 ± 0.11 | 1.22 ± 0.21 | 1.13 ± 0.16 |
| APOE-ε4 n alleles(1%,2%) | 83(28)/16(5) | 157(47)/18(5) | 240(38)/34(5) | 182(25)/21(3) |
| TREM2 Gene Burden n(%) | 17(5.7) | 23(6.9) | 40(6.4) | Not Available |
| rs2234256 n | 7 | 4 | 11 | Not Sampled |
| rs2234255 n | 0 | 1 | 1 | Not Sampled |
| rs142232675 n | 4 | 0 | 4 | 2 |
| rs143332484 n | 7 | 13 | 20 | 13 |
| rs75932628 n | Not Sampled | 5 | 5 | Not Sampled |
| rs2234253 n | Not Sampled | 4 | 4 | Not Sampled |

The discovery sample was comprised of individuals from ADNI and A4 with multimodal PET imaging and whole genome sequencing. The replication sample was taken from the HABS-HD dataset.
n sample size, CN cognitively normal, MCI mild cognitive impairment, AD Alzheimer's disease, Aβ amyloid β, EC Tau SUVR entorhinal cortex tau standardised uptake value ratio, MetaTemp Tau SUVR temporal meta region of interest tau standardised uptake value ratio, APOE-ε4 Apolipoprotein E -ε4, TREM2 triggering receptor expressed on myeloid cells 2.
[a]4 ADNI participants were missing diagnosis.
[b]11 ADNI participants were missing race.
[c]4 A4 participants missing race.
[d]15 participants in the combined sample were missing race.
[e]The discovery Sample used FTP tau PET imaging, whereas the Replication sample used PI2620 tau PET imaging.

there will be variable and interactive effects of TREM2 and APOE-ε4 that increase tau spreading from primary regions into the neocortex (i.e. greater neocortical tau for a given level of primary tau).

## Results

### Participants
We pooled data from the Alzheimer's Disease Neuroimaging Initiative (ADNI) and Anti-Amyloid Treatment in Asymptomatic Alzheimer's Disease (A4) study as a discovery sample ($n = 628$, 79% cognitively normal) and drew an ethnically and racially diverse replication sample ($n = 726$, 76% cognitively normal) from the Health and Aging Brain Study-Health Disparities (HABS-HD) cohort. Participants had varying levels of Aβ, EC tau and neocortical tau as defined using the meta-temporal (MetaTemp) region of interest (ROI)[36]. Furthermore, we extracted genetic information from participants assessing the number of APOE-ε4 alleles in both samples, as well as a binary indicator for TREM2 risk variant carrier status in the discovery sample. Due to the limited TREM2 single nucleotide polymorphism (SNP)s data in the HABS-HD, we omitted this variable from our replication analysis (Table 1).

### Effects of different genotypes on Aβ
We first assessed which genetic variants affected levels of Aβ while including age as a confounding variable. Within the discovery sample, we observed significant effects of age ($\beta = 0.9650$ $p < 0.0001$) and APOE-ε4 (1 allele vs 0:$\beta = 27.7510$ $p < 0.0001$; 2 allele vs 0:$\beta = 41.9344$ $p < 0.0001$). We observed no differences in Aβ contrasting sex (Female vs Male: $\beta = -1.2392$ $p = 0.6860$) or TREM2 risk variant carrier status (1 vs 0: $\beta = 0.0734$ $p = 0.9905$) (Supplementary Table 1). Similar results were observed in the HABS-HD sample, with significant effects of age ($\beta = 1.16$ $p < 0.0001$) and APOE-ε4 (1 allele vs 0:$\beta = 11.90$ $p < 0.0001$; 2 allele vs 0:$\beta = 24.46$ $p < 0.0001$). However, we also observed a significant sex effect in this sample (Female vs Male: $\beta = 5.58$ $p = 0.004$) (Supplementary Table 2). We further tested if self-reported race confounded these results in the HABS-HD dataset including race as a main effect and interaction term with genetic variables. Both APOE-ε4 and

sex remained significant in their effects on Aβ, however their effect sizes differed across different racial ethnic groups (Supplementary Table 3). Finally, we observed negligible differences in estimated parameters for each sample when including diagnosis as a confounding variable (Supplementary Tables 4, 5).

### Effects of different genotypes on EC-tau
Next, we built a causal path model (Fig. 2f) to test the main and interactive effects of different genotypes on EC tau (Methods EQ2). Within the discovery sample, we observed significant main effects of Aβ ($\beta = 0.0013$, $P < 0.0001$) and age ($\beta = 0.003$, $P = 0.0072$) on EC tau. Furthermore, we observed significant interactions between Aβ and APOE-ε4(2-alleles) ($\beta = 0.002$, $P = 0.002$), and Aβ and sex(Female) ($\beta = 0.0008$, $p = 0.032$). Visualising the marginal effects indicate that, for a given level of Aβ, APOE-ε4 homozygotes and females had significantly more tau in the EC (Fig. 2a, b, Supplementary Fig. 1). In addition, we observed a significant interaction between TREM2 and sex, whereby female TREM2 risk variant carriers had the greatest levels of EC tau ($\beta = 0.124$ $p = 0.0407$) (Supplementary Table 6). We observed highly similar interaction terms in the HABS-HD sample, observing Aβ interacts with APOE-ε4(2-alleles) ($\beta = 0.0037$, $P = 0.018$) and sex(Female) ($\beta = 0.0013$, $p = 0.032$) (Supplementary Table 7). These interactions replicate results showing that for a given level of Aβ, APOE-ε4 homozygotes and females have significantly more tau in the EC (Fig. 2d, e, Supplementary Fig. 1). Our results are likely not confounded by race in the replication sample, as when self-reported race was included as a main effect and interaction term with genetic variables in our model, none of these added variables had a significant influence on EC tau (Supplementary Table 8). Finally, we observed negligible differences in estimated parameters for each sample when including diagnosis as a confounding variable (Supplementary Tables 9, 10).

### Effects of different genotypes on MetaTemp tau
We then built a causal path model (Fig. 3e) to test the main and interactive effects of different genotypes on MetaTemp tau (Methods

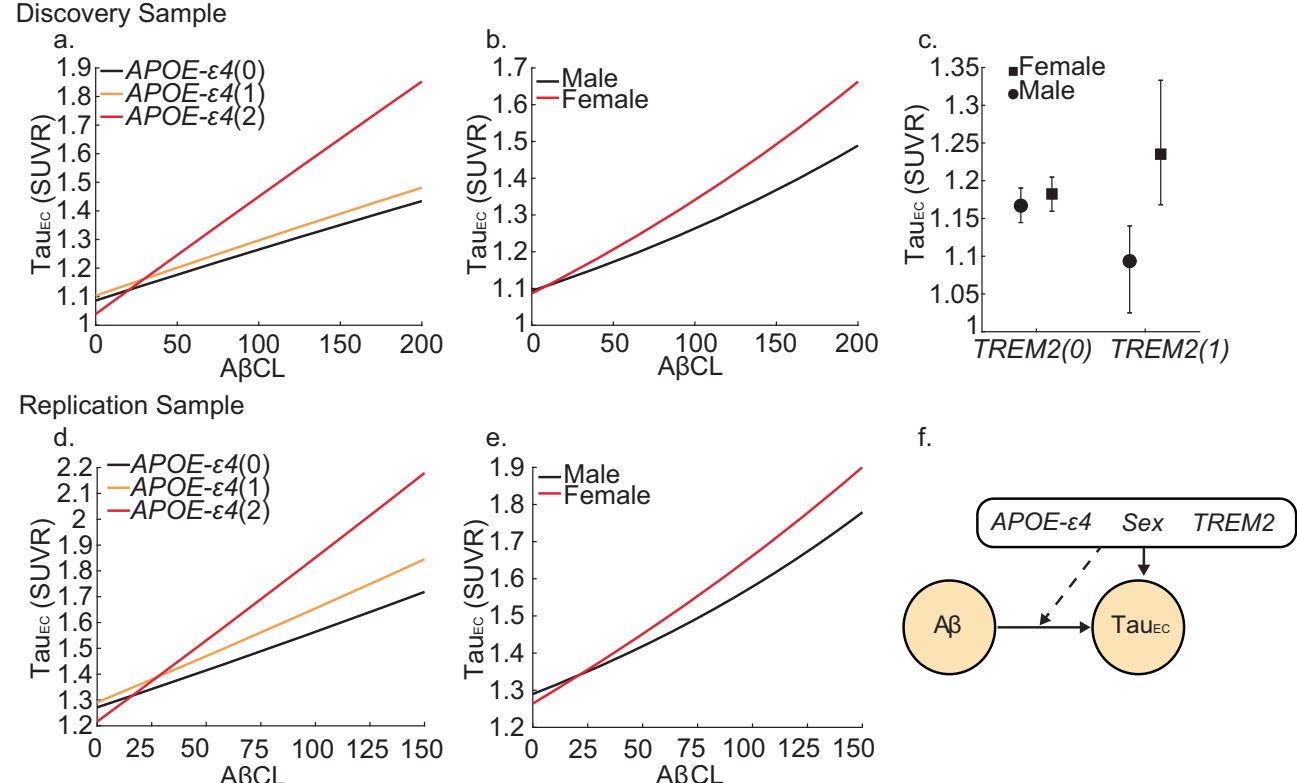

**Fig. 2 | Genetic influences on entorhinal cortex tau.** Lines show the estimated marginal levels of tau pathology for individuals with varying genetic profiles at different levels of amyloid beta (Aβ). Whisker plot in panel c shows estimated levels of entorhinal cortex tau (Tau$_{EC}$) pathology based on sex and Triggering Receptor Expressed on Myeloid Cells 2 (*TREM2*) risk variant carrier status, marginalised over Aβ. Points represent estimated Tau$_{EC}$ pathology derived from multiple linear regressions, and error bars indicate 95% two-sided confidence intervals computed from 1000 multiplier bootstrap replications. Different effects in the discovery sample ($n = 626$) for (**a**) Apolipoprotein E (*APOE*) -*ε4*, (**b**) sex, (**c**) interaction of sex

and *TREM2* risk variant carrier status. Different effects in the replication sample ($n = 726$) for (**d**) *APOE-ε4* and (**e**) sex. **f** Causal path model to estimate Tau$_{EC}$. Solid lines indicate a direct effect of an upstream variable on a downstream pathology, dashed lines indicate an interactive effect between upstream variables (i.e. Aβ centiloid or genetic variable) on Tau$_{EC}$. Lines initiating from the black genetic variables box may indicate a direct or interactive effect of genetic variables on the outcome variable. Values in the parentheses indicate level of genetic variable (*APOE-ε4* alleles (0,1,2) and *TREM2* risk variant carrier status (0,1)). Source data are provided as a Source Data file.

EQ3). To allow for an accurate estimation of the effects when two continuous variables interact, we segmented Aβ centiloid (CL) values into four bins (<10CL, 10-40CL, 40-60CL and >60CL) when including it as an interactive term with EC tau. This Aβ variable is then used as a candidate variable when selecting interaction terms for the causal path model for MetaTemp tau. Continuous Aβ is still included as a main effect variable in model selection. Within the discovery sample, we observed significant main effects of EC tau (β = 0.681, $P < 0.001$), *TREM2* risk variant carrier status (β = −0.358 $p < 0.001$) and *APOE-ε4* homozygosity (β = −0.464 $p < 0.001$) on MetaTemp tau. In addition, we observed significant interactions between EC tau and *APOE-ε4*(2-alleles) (β = 0.364, $P < 0.001$), EC tau and *TREM2*(1) (β = 0.309, $P < 0.001$), and EC tau and high Aβ (> 60CL) (β = 0.019 $p = 0.049$) although the latter is a small effect. Individuals who are *APOE-ε4* homozygotes and individuals with a *TREM2* risk variant carrier status, and to a lesser extent, individuals with high levels of Aβ had greater downstream MetaTemp tau burden for a given level of EC tau (Fig. 3a, b, Supplementary Fig. 2). A significant interaction between *APOE-ε4* and *TREM2* (β = 0.0821, $p = 0.0168$) indicated that those with a *TREM2* risk variant and an *APOE-ε4* allele had significantly more MetaTemp tau. Visualising the marginal effects highlights these results showing that *APOE-ε4* homozygotes and *TREM2* risk variant carriers have greater MetaTemp tau, and these differences increase with EC tau burden, independent of Aβ main and interactive effects with EC tau (Fig. 3c, Supplementary Fig. 2). We did not observe any significant effects of sex on MetaTemp tau (sex(Female) (β = 0.0613, $p = 0.1407$);

sex(Female)*EC tau (β = −0.0504, $p = 0.1496$)) (Supplementary Table 11). Where possible due to available data, we observed highly similar results in the HABS-HD sample. We observed significant effects of *APOE-ε4* homozygosity on MetaTemp tau both as a main effect (β = −0.217 $p = 0.0069$) and interacting with EC tau burden (β = 0.157, $P = 0.0047$) (Fig. 3d, Supplementary Fig. 2). This replicates the association we observed showing that *APOE-ε4* homozygotes have greater MetaTemp tau, and these differences increase with EC tau burden independent of Aβ effects. Similar to the discovery sample, we did not observe any significant effects of sex on MetaTemp tau (sex(Female) (β = −0.013, $p = 0.74$); sex(Female)*EC-tau (β = 0.0305, $p = 0.31$)) (Supplementary Table 12). Like the previous analysis for EC tau, our results are likely not confounded by race in the replication sample. When self-reported race was included as a main effect and interaction term with genetic variables in our model, these added variables did not have significant influences on MetaTemp tau (Supplementary Table 13). Furthermore, we observed negligible differences in estimated parameters for each sample when including diagnosis as a confounding variable (Supplementary Tables 14, 15).

## Variable genetic effects at different stages in the canonical amyloid cascade pathway

Finally, we investigated the variability in the direct, mediation, and total effects of upstream pathology on downstream tau pathology for different genetic profiles. This allows us to probe each aspect of the AD cascade and estimate which groups are likely to have higher downstream

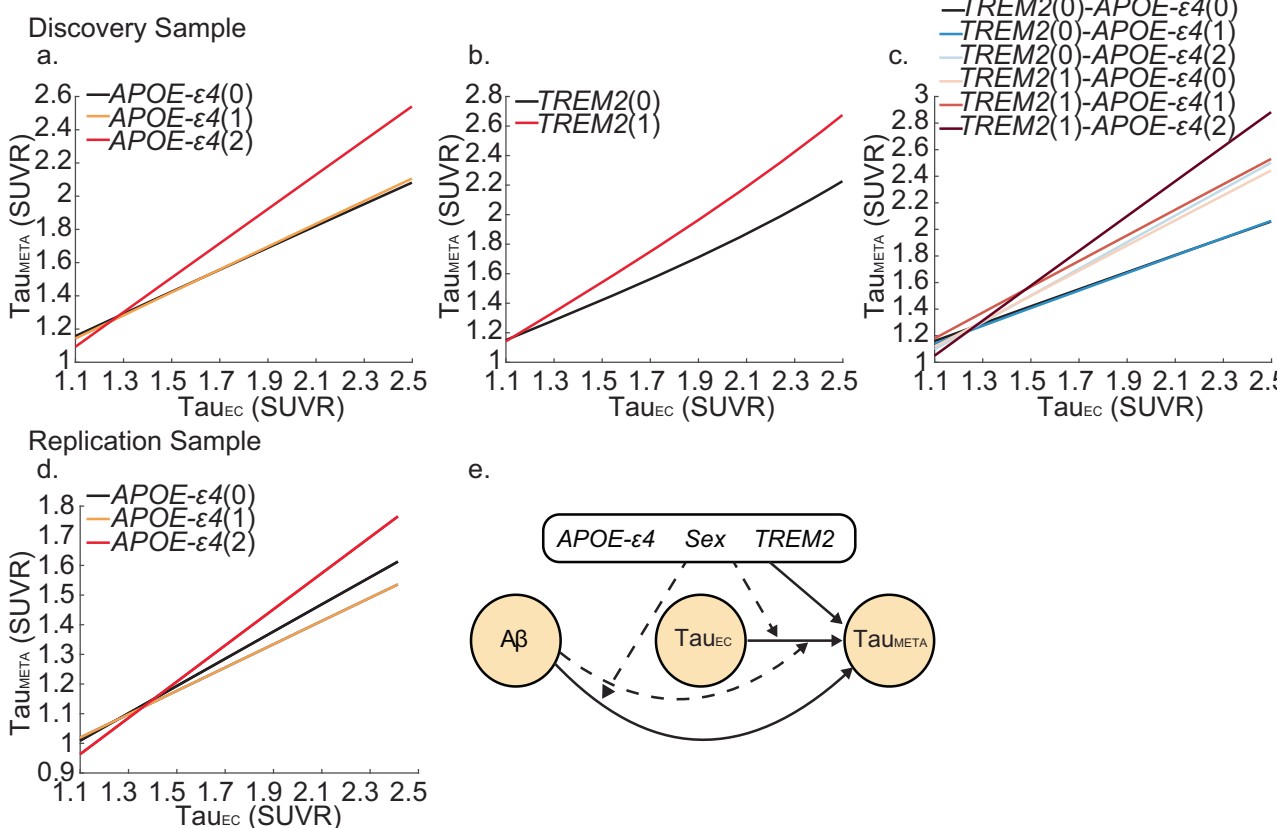

**Fig. 3 | Genetic influences on meta-temporal tau.** Lines show the estimated marginal levels of meta-temporal tau (Tau$_{Meta}$) pathology for individuals with varying genetic profiles at different levels of entorhinal cortex tau (Tau$_{EC}$), with amyloid beta (Aβ) fixed at 40 centiloids to facilitate comparison. Different effects in the discovery sample (n = 626) for (**a**) Apolipoprotein E (*APOE*)-*ε4*, (**b**) Triggering Receptor Expressed on Myeloid Cells 2 (*TREM2*), and **c**. their interaction. Different effects in the replication sample (*n* = 726) for (**d**) *APOE-ε4*. **e** Causal path model to estimate Tau$_{Meta}$. Solid lines indicate a direct effect of an upstream variable on a downstream pathology, dashed lines indicate an interactive effect between upstream variables (i.e. Aβ, Tau$_{EC}$, or genetic variable) on meta-temporal tau. Lines initiating from the black genetic variables box may indicate a direct or interactive effect of genetic variables on the outcome variable. Values in the parentheses indicate level of genetic variable (*APOE-ε4* alleles (0,1,2) and *TREM2* risk variant carrier status (0,1)). Source data are provided as a Source Data file.

pathology for a given level of upstream pathology (i.e. if a population has greater EC tau for a given level of Aβ and how this may result in higher levels of MetaTemp tau). To do this, in each group we calculate the direct effect of Aβ on EC tau (DE$_{Aβ ->EC\ tau}$ Fig. 4a, e); the direct effect of EC tau on MetaTemp tau (DE$_{EC\ tau ->MetaTemp\ tau}$ Fig. 4b, f); the direct effect of Aβ on MetaTemp tau (DE$_{Aβ ->MetaTemp\ tau}$); the mediation effect of Aβ through EC tau on MetaTemp tau (ME$_{Aβ ->EC-tau -> MetaTemp\ tau}$ Fig. 4c, g), which is the product between the direct effect of EC tau on MetaTemp tau and the direct effect of Aβ on EC tau; and finally the total effect of Aβ on Meta-Temp tau (TE$_{Aβ -> MetaTemp\ tau}$ Fig. 4d, h), which is calculated as the direct sum of the mediation effect through the path ME$_{Aβ ->EC-tau -> MetaTemp\ tau}$ and the direct effect of Aβ on MetaTemp tau. Since Aβ does not interact with genetic factors in the MetaTemp Tau response model, the direct effect of Aβ on MetaTemp Tau is a constant across different genetic groups. Consequently, the effect difference among these groups will always be zero.

In the discovery sample we observed that the total effect of Aβ on MetaTemp tau differs by APOE-ε4 allele status. In particular, APOE-ε4 homozygotes have significantly higher MetaTemp tau levels at a given level of Aβ compared to individuals with zero or one ε4 allele (TE$_{Aβ -> MetaTemp\ tau}$: APOE-ε4 (2 alleles) vs. (0 alleles)=0.003, *p* = 0.022; APOE-ε4 (2 alleles) vs. (1 allele)=0.0029, *p* = 0.035; see Fig. 4d, Supplementary Table 16). This difference is driven by the mediation pathway from Aβ to EC tau and subsequently to MetaTemp tau (Fig. 4c). Because the mediation effect through this pathway is the product of the direct effect of Aβ on EC tau and the direct effect of EC

tau on MetaTemp tau (Fig. 4a, b), these findings imply that in APOE-ε4 homozygotes, a given level of Aβ leads to higher EC tau, and a given level of EC tau leads to higher MetaTemp tau. Consequently, this population shows elevated MetaTemp tau at a given level of Aβ due to enhanced Aβ-driven EC tau aggregation and enhanced EC tau propagation into the neocortex.

There were no significant differences in the TE$_{Aβ -> MetaTemp\ tau}$ (Fig. 4d) across other genetic groups, but we note that several numerical differences were present (albeit not reliably statistically significant). In the TREM2 group, the TE$_{Aβ -> MetaTemp\ tau}$ (Fig. 4d) was not significant; however, the mediation effect through Aβ and EC tau (i.e., Aβ → EC tau → MetaTemp tau) was significant (*TREM2*(1) vs *TREM2*(0) = 0.0008, *p* = 0.014) (Fig. 4c). This was driven by significant differences in the direct effect of Aβ on EC tau (*TREM2*(1) vs *TREM2*(0) = 0.0002, *p* = 0.027) (Fig. 4a) and in the direct effect of EC tau on MetaTemp tau (*TREM2*(1) vs *TREM2*(0) = 0.3145, *p* = 0.029) (Fig. 4b) (Supplementary Table 16). These findings suggest that, in *TREM2* risk variant carriers, across levels of Aβ there are higher levels of EC tau, and in turn, a given level of EC tau leads to higher MetaTemp tau. Although the mediation effect is significant, the overall total effect is diminished by the low-powered direct effect of Aβ on MetaTemp tau, likely due to the limited sample size of *TREM2* risk variant carriers in the discovery cohort.

For females vs males comparison there was a non-significant numerical difference in the TE$_{Aβ -> MetaTemp\ tau}$ (females vs males = 0.0005, *p* = 0.12) (Fig. 4d), which was driven by significant differences

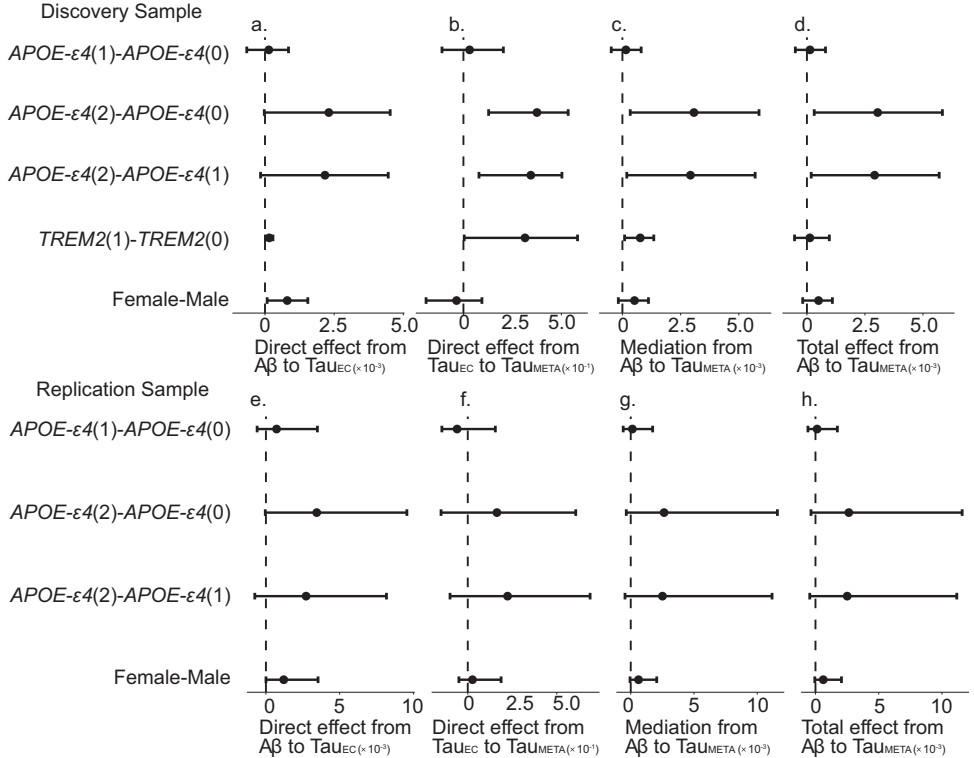

**Fig. 4 | Direct, mediation and total effects of upstream pathology on tau for different levels of sex, *APOE-ε4*, and *TREM2* risk carrier status.** Estimated effects along the amyloid beta (Aβ) → entorhinal cortex tau (Tau$_{EC}$) → meta-temporal tau (Tau$_{Meta}$) cascade are shown for contrasts defined by Apolipoprotein E (*APOE*)-*ε4* alleles (0,1,2), Sex (female, male), and Triggering Receptor Expressed on Myeloid Cells 2 (*TREM2)* risk variant carrier status (0,1). The top row (a–d) shows results from the discovery sample ($n = 626$), and the bottom row (e–h) shows results from the replication sample ($n = 726$). Panels (**a**) and (**e**) show the direct effect from Aβ to Tau$_{EC}$; panels (**b**) and (**f**) show the direct effect from Tau$_{EC}$ to Tau$_{Meta}$; panels (**c**) and

(**g**) show the mediation effect of Aβ on Tau$_{Meta}$ via Tau$_{EC}$; and panels (**d**) and (**h**) show the total effect of Aβ on TauMeta. Each point represents the estimated effect contrast between two levels (e.g., APOE-ε4(2) - APOE-ε4(0) corresponds to the effect estimate for APOE-ε4 homozygotes minus that for non-carriers), and error bars indicate confidence intervals derived from 1000 multiplier bootstrapped samples: 95% two-sided for the discovery sample (**a**–**d**), and 95% one-sided (5th to 100th percentile) for validation in the replication sample (**e**–**h**). Source data are provided as a Source Data file.

in the DE$_{Aβ \to EC\ tau}$ (females vs males=0.0008, $p = 0.036$) (Fig. 4a), (Supplementary Table 16). This implies that for a given level of Aβ, women have greater levels of EC tau, and a trend towards greater MetaTemp tau for the same level of Aβ, a finding which was supported in the HABS-HD replication sample. Relative effects and interpretations for other contrasts were similar in the replication sample (Supplementary Table 17, Fig. 4e–h).

## Discussion

Here we used causal path analyses structured on the canonical AD pathological cascade where Aβ is the initiating event, followed by increased tau burden in the EC, followed by tau involvement of neocortex. Using this path framework, and genetic variations in sex, *TREM2* and *APOE-ε4* as instrumental variables we can infer how different biological processes influence the AD cascade leading to increases in discrete stages of tau pathology (i.e., medial temporal tau and neocortical tau). We provide compelling evidence for heterogeneity in how regionally specific tau pathology is distributed based on different genetic traits (Fig. 5), thus providing insight into the biological mechanisms that may govern increased tau pathology. Furthermore, we provide empirical evidence that may explain variable gene and sex related treatment effects of recent anti-amyloid immunotherapy trials.

We observe strong effects of *APOE-ε4* homozygosity on both tau in EC and neocortex (Fig. 5a). Specifically, across levels of Aβ *APOE-ε4* homozygotes had substantially more EC tau. Further, for a given level of EC tau *APOE-ε4* homozygotes had greater levels of neocortical tau

after accounting for both direct and interactive effects of EC tau and Aβ. The net result of these effects means in comparison to *APOE-ε4* non-carriers and heterozygotes *APOE-ε4* homozygotes have a greater level of EC tau for a given level of Aβ, and a greater level of neocortical tau for a given level of EC tau.

A notable result in this work is the relative absence of tau related effects for *APOE-ε4* heterozygotes. We observed a strong and reliable dose dependent effect of the number of *APOE-ε4* alleles on Aβ burden, with subsequent downstream modelling showing effects of Aβ on EC tau as well as interactive effects of Aβ and EC tau on MetaTemp tau. It is likely through these indirect Aβ mediated routes the number of *APOE-ε4* alleles has a dose dependent effect on tau. Our path modelling approach however conditions downstream models (i.e. with tau as an outcome) in a way to account for these Aβ mediated dose dependent effects, with the resultant effects of *APOE-ε4* homozygosity on tau over and above these Aβ mediated effects. Our analyses suggest that the sole presence of the APOE4 isoform in humans has a strong effect on tau, both through interactions with existing Aβ pathology and interactions with primary tau pathology.

Multiple lines of evidence have shown that overexpression of APOE4 increases tau phosphorylation[14,37] and spread of tau[9,12]. In knock in mouse models with human *APOE-ε4* homozygosity it is clear that only expressing the APOE4 isoform increases tau pathology through the activity of neuronal[38,39], astrocytic[40] and microglial[41,42] cells. Prior work in these mouse models with blocked expression of APOE4 in neuronal or astrocytic cells resulted in a reduction of tau phosphorylation and spread compared to the mice solely expressing APOE4.

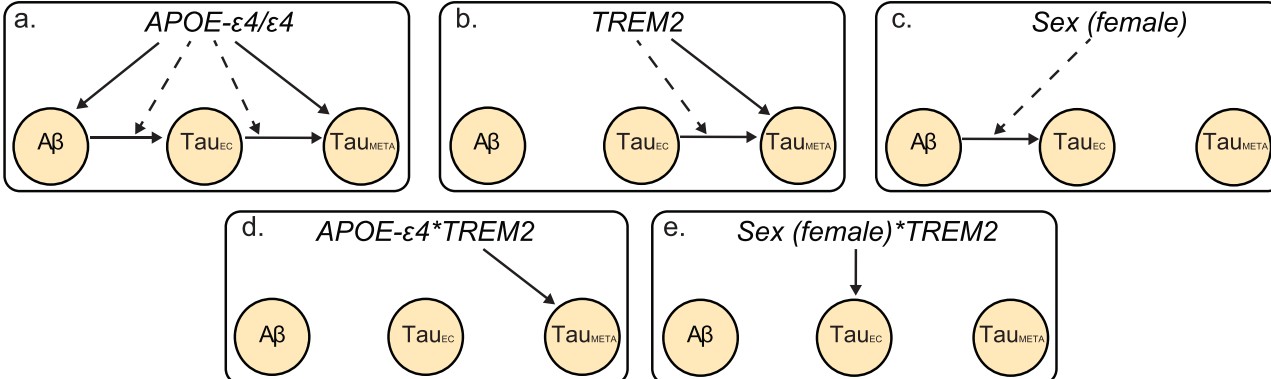

**Fig. 5 | Genetic influences on the AD cascade.** Causal path models indicating significant pathways in which genetic variation leads to increased Alzheimer's Disease (AD) pathology. Each line originates at a genetic variable with the yellow nodes indicating an outcome variable. Solid lines indicate the pathway from an upstream variable to a downstream pathology, dashed lines indicate an interactive effect between upstream variables on a downstream pathology, with the arrow indicating a modulation of the pathway shown by the solid line. **a** effect of Apolipoprotein E (*APOE*)-ε4 homozygosity on AD pathology. **b** Effect of Triggering Receptor Expressed on Myeloid Cells 2 (*TREM2*) risk variant carriership on AD pathology. **c** effect of female sex on AD pathology. **d** interactive effect of *APOE*-ε4 and *TREM2* risk variant carrier status on AD pathology. **e** Effect of female sex and *TREM2* risk variant carrier status on AD pathology.

Indicating that in the presence of only APOE4, tau proliferates to a greater extent. Furthermore, *APOE*-ε4 knock in mice have displayed hyperexcitability in medial temporal regions[39,43], a potential driver of tau accumulation[44].

Converging human neuroimaging studies have implicated the *APOE*-ε4 allele with increased levels of medial temporal tau independent of Aβ[45–48], although some reports do indicate that Aβ may mediate the relationship between *APOE*-ε4 and medial temporal tau. A likely influence of these discrepant results is the aggregation of *APOE*-ε4 carriership as a binary variable and not modelling *APOE*-ε4 homozygotes as a different population. Given the substantially greater risk of having clinical AD[6] and Aβ positivity[7] in homozygotes compared to heterozygotes, it is becoming clear that the statistical aggregation of *APOE*-ε4 homozygotes and heterozygotes in one group may not be appropriate[7]. Although we were well powered to make comparisons between *APOE*-ε4 homozygotes and heterozygotes, further work in larger samples will be required to fully understand the differences in tau burden between these groups.

Recent imaging studies have suggested that *APOE*-ε4 potentiates the relationship between Aβ and tau pathologies[49,50]; our results provide some support of this model, highlighting that Aβ and *APOE*-ε4 homozygosity interact to increase EC tau pathology. However, we suggest a refinement to this model whereby *APOE*-ε4 homozygosity also potentiates the relationship between tau burden in the EC and spread into the neocortex through interactions with tau independent of Aβ.

We have shown that *APOE*-ε4 homozygotes have increased primary tau pathology in the EC at lower levels of Aβ and increased neocortical tau pathology at lower levels of EC tau. This strongly suggests that homozygotes should be treated earlier with anti-amyloid treatment (i.e. at lower levels of Aβ) to reduce tau, not simply because they begin depositing Aβ earlier than other genotypes[51] but because this Aβ drives higher tau. In addition, the effects of homozygosity on increased neocortical tau relative to EC tau raise questions about the need for anti-tau therapy in this group. This empirical evidence may provide some of the biological underpinnings explaining the lack of treatment effect in *APOE*-ε4 homozygotes in both the Lecanemab and Donenamab trials[34,35].

Using rare polymorphisms on the coding region of the *TREM2* gene, we find that trait differences in the function of TREM2 plays a role in the spread of tau from the EC into the neocortex (Fig. 5b). Using causal path modelling we were able to infer at which stage genetic variation on the *TREM2* gene may become relevant in the AD cascade

(i.e. before/after Aβ accumulation, or before/after EC tau accumulation). We observed a strong effect showing that for a given level of tau in the EC, individuals with a gene burden of *TREM2* have greater levels of neocortical tau after accounting for other upstream and confounding variables (i.e. sex, *APOE*-ε4, age, Aβ). The function of TREM2 has been linked to tau spread through several putative mechanisms[16,17]. In particular, animal models have highlighted that *TREM2* risk variants result in hypoactive TREM2 which may disrupt microglial inflammatory signalling to tau and may promote tau transmission through aberrant microglial activity[22,52–55]. Prior neuroimaging studies in humans has similarly implicated microglial activity in tau deposition in the neocortex[56]. Furthermore, CSF soluble TREM2 (sTREM2) levels are predictive of the transition from pre-clinical to clinical AD and are associated with CSF tau levels[16]. Alternate accounts have also shown that increased CSF levels of sTREM2 may be protective against tau and Aβ accumulation[57]. Our work adds to this body of literature by interrogating the function of TREM2 as a parent trait variable rather than an indicator which varies as a function of pathological state, providing in vivo evidence into how dysfunction in TREM2 enters the AD cascade, working to exacerbate tau spread from the EC into the neocortex likely through aberrant microglial function. Incorporating sTREM2 in a similar path modelling framework as a microglia response phenotype downstream to both genetic risk of *TREM2* and Aβ may provide additional state related changes in TREM2 further explaining the role of microglia in response to, and, in driving AD pathologies.

In addition to the independent effects of *TREM2* and *APOE*-ε4, we observed a significant interaction between these two genetic risk traits. Individuals with a *TREM2* risk variant and an *APOE*-ε4 allele had higher levels of neocortical tau (Fig. 5d), implicating interactive factors between APOE4 and TREM2 on tau aggregation. APOE is a ligand of TREM2, with models of *TREM2* variants (R47H, R62H, D87N) showing a decreased binding affinity between TREM2 and APOE in vitro[17,58–60]. Furthermore, prior work shows that APOE4 isoforms are recognised and engulfed by TREM2 at different rates than other APOE isoforms[61]; this differential binding of TREM2 to APOE4 may result in an impaired switch of homoeostatic microglia to disease-associated microglia in AD[16,18,19]. Previous neuroimaging studies have also implicated the *APOE*-ε4 allele as a modulating factor between microglial activity and tau spread independent of Aβ[45]. Our results support this previous human work highlighting that genetic traits that may manifest in aberrant interactions between TREM2 and APOE4 likely have a profound effect on the spreading of tau from the EC into the neocortex.

When investigating how sex interacts with Aβ we observe that females have higher levels of EC tau for a given level of Aβ than their male counterparts (Fig. 5c). This result fits well with previous accounts showing females have higher levels of medial temporal lobe tau after controlling for Aβ[30–32]. We did not observe significant effects of sex on tau in neocortical regions suggesting that females are more susceptible to early tau deposition in the EC but may not show differences in tau spreading into the neocortex. This suggests that females should see similar benefits in anti-amyloid treatment as males but may require treatment at lower levels of Aβ --so as to be at similar levels of primary tau-- or to be screened for both Aβ and tau, ensuring that females do not have more advanced tau pathology than their male counterparts. These findings may also provide some insights into the discrepancies between the Lecanemab and Donenanab trials, whereby females did not see the same benefit as males in Lecanemab but did in Donenanab[34,35]. It is feasible that the multimodal screening for intermediate tau and Aβ positivity in Donenanab vs. unimodal screening of Aβ positivity in Lecanemab ensured that the sex related differences in early tau burden were ameliorated. Confidence in such an interpretation will require further exploration. After accounting for upstream pathologies, we observed that females who harbour a risk variant in *TREM2* have higher levels of EC tau after accounting for all other upstream or confounding variables (i.e. Aβ, *APOE-ε4* and age) (Fig. 5e). Previous work has pointed to interactive factors of microglial activity and sex, implicating a greater role of microglia in tauopathy for females[33]. Here we show similar trait interactions between the risk gene burden of *TREM2* and female sex in EC tau after accounting for upstream variables.

Our interpretation and approach have several limitations. First, we note that our interpretation of our results in light of the recent anti-amyloid trials is highly susceptible to confirmation bias. Although we initiated this work prior to the release of the results of these trials, we did undertake much of the preparation of this manuscript knowing that *APOE-ε4* homozygotes and females may have variable treatment outcomes. Although our results fit well in existing literature, further work is required to understand the reasons that these populations saw attenuated or no benefit in the respective trials. Second, although our model is grounded in generally well accepted neuropathological staging of AD[62,63], it is cross sectional so further modelling work on longitudinal multimodal imaging datasets is required to fully elucidate the dynamics of AD pathophysiological changes. Third, our interpretation of the interactive effects of EC tau and *TREM2, or APOE-ε4* homozygosity on neocortical tau assumes typical Braak like progression (i.e. tau spreading from EC into neocortex), however it is plausible that our results could be consistent with a more atypical non-Braak like presentation of tau (i.e. high neocortical tau but no EC tau). Previous neuropathological work in patients with varied AD phenotypes suggests that *TREM2* variant carriers have a higher proportion of atypical, hippocampal sparing patterns of tau burden[64]. However, the observation that in *TREM2* risk variant carriers there is still a positive interaction between EC-tau and MetaTemp tau levels suggests that the accumulation of tau in the neocortex is not entirely decoupled from the medial temporal lobes (i.e. hippocampal sparing). Fourth, our analytical decision to consolidate multiple *TREM2* risk variants into a risk variant carrier status assumes a similar biological role of each risk variant and does not apply a weighting to individual SNPs[65]. Furthermore, the 6 SNPs were not fully sampled in each cohort with 4 of 6 sampled in ADNI and some variants were imputed from the WGS in A4. Therefore, it is possible that some individuals who have a *TREM2* risk variant were assigned to the non-carrier group due to a lack of available data. However, both of these analytical decisions would work to push our results closer to the null (i.e. more prone to type II error) and thus do not negate the findings presented here. Future work on larger and more complete datasets will afford a more granular appraisal of the biological role of each risk SNP on the AD cascade.

Similarly, although we validated our findings related to sex and *APOE-ε4* in an ancestrally diverse sample, we were unable to undertake robust ancestral stratification to fully understand potential ancestral differences that are observed in AD genetic risk[6]. However, we do note that there was a reasonable representation of each ethnicity in our validation sample of *APOE-ε4* homozygotes. Further, although our sample is large, the number of *APOE-ε4* homozygotes is still small. However, we did replicate our findings in two independent and relatively heterogeneous samples, giving further confidence to our findings regarding *APOE-ε4*. Finally, we were unable to run a replication analysis of TREM2 effects in the HABS-HD sample. This was due to the incomplete sampling of *TREM2* SNPs, limiting our ability to generate a reliable *TREM2* risk variant carrier status resulting in a substantially lower proportion of *TREM2* risk variant carriers than the discovery sample. This low proportion resulted in no *TREM2* risk variant carriers who were *APOE-ε4* homozygotes thus restricting our ability to appropriately model the data. Further work will be required to validate our *TREM2* findings, particularly those close to significance thresholds (i.e. interactions with sex and *APOE-ε4*). Notwithstanding, our findings regarding the main effects and interactions of *TREM2* with EC tau on neocortical tau were well below our significance threshold providing some confidence in these findings.

Here, we have used genetic variation to provide a deeper understanding of the biological mechanisms that drive AD pathophysiology. We show in a diverse sample of over 1300 participants that females and *APOE-ε4* homozygotes are more susceptible to the primary accumulation of tau, with greater EC tau for a given level of Aβ. Furthermore, we observed for individuals with risk variants in *TREM2* and *APOE-ε4* homozygotes there was a greater spread of primary tau from the EC into the neocortex. These findings offer insights into the function of sex, APOE and microglia (vis a vis TREM2) in AD progression and have implications for determining personalised treatment with drugs targeting Aβ and tau.

## Methods
### Participants
This study was approved by the Institutional Review Board (IRB) of the University of California, Berkeley. The A4 Study was conducted under ethical approvals obtained from the IRBs of each participating site, coordinated by the study leadership. Participants provided written informed consent after a comprehensive explanation of the study procedures, including PET imaging, cognitive assessments, and data sharing for research purposes. The HABS-HD study was approved by the IRBs of the University of North Texas Health Science Center. All participants provided written informed consent, allowing for the collection and use of imaging, cognitive, genetic, and demographic data in research. All ADNI participants provided written informed consent, and the study protocols were approved by the IRBs of all participating institutions.

We pooled two well-characterised ageing and AD cohorts as a discovery sample (*n* = 628, 79% cognitively normal), two participants were quantitatively determined as multivariate outliers and were removed from the sample. A sample of clinically impaired and cognitively normal (*n* = 297, 55% cognitively normal) participants were drawn from ADNI and were combined with a sample of cognitively normal (*n* = 331) participants drawn from the screening visit of the A4 study. We selected participants from their respective studies who had whole genome sequencing (WGS), tau and Aβ PET imaging. This restricted the A4 sample to predominantly represent the elevated Aβ group with a small subset from the Aβ not elevated LEARN observational study. The majority of participants within this discovery sample were self-described as White (93%). In addition, we drew an ethnically and racially diverse replication sample (n = 726) from the HABS-HD cohort, selecting participants who had tau, Aβ PET imaging, and *APOE-ε4* genotyping. This sample was predominantly cognitively

normal (76%) and well balanced for self-reported racial/ethnic diversity amongst White (38%), Hispanic (30%) and Black (32%) populations (Table 1).

## Neuroimaging

**Flortaucipir (FTP)- PET Tau.** The ADNI [18F] Flortaucipir (FTP) FTP-PET protocol entailed the injection of 10 mCi of FTP followed by acquisition of 30 min of emission data from 75 to 105 min post-injection. The A4 FTP-PET protocol acquired 30 min of emission data from 80 to 110 min post-injection. The HABS-HD [18F] PI-2620 tau PET protocol entailed the injection of 10 mCi of PI-2620 followed by the acquisition of 30 min of emission data from 45–75 min.

Tau PET data were realigned, and the mean of all frames was used to coregister tau PET to each participant's MRI acquired closest to the time of the tau PET. Tau PET standardised uptake value ratio (SUVR) images were normalised to inferior cerebellar grey matter. MR images were segmented and parcellated into the Desikan-Killiany atlas using Freesurfer (V5.3) and regions of interest were used to extract cerebellar-normalised regional SUVR data. SUVR data was summarised for two regions of interest (ROIs) in the entorhinal cortex and the tau meta-temporal (MetaTemp) ROI, comprised of the volume weighted average of the entorhinal, amygdala, parahippocampal, fusiform, inferior temporal, and middle temporal ROIs. Due to variable scanner resolution in the A4 FTP dataset, the smoothness of data for each scan site could not be reliably estimated for all subjects, therefore we did not partial volume correct any dataset. All tau PET data were analysed using the same in-house MATLAB R2023a pipeline at UC Berkeley and SUVR values are used throughout as a tracer specific harmonised scale for EC tau and MetaTemp tau.

**Aβ PET imaging.** ADNI Aβ imaging was performed at each ADNI site using either [18F] Florbetapir (FBP) or [18F] Florbetaben (FBB). A4 Aβ imaging was performed at each site using FBP. HABS-HD Aβ imaging was performed using FBB. The FBP Aβ-PET protocol involved injection of 10 mCi of FBP followed by the acquisition of 20 min of emission data at 50–70 min post-injection. The FBB protocol involved injection of 8.1 mCi of FBB followed by 20 min of emission data at 90–110 min post-injection. Aβ-PET images were then processed to derive a summary of global Aβ burden in centiloids (CL) that are used throughout as a harmonised scale for Aβ-PET burden[66]. The CL conversion for the ADNI and HABS-HD samples was performed using in-house MATLAB R2023a processing pipelines at UC Berkeley, A4 CL values were downloaded from LONI.

## Genetics

**TREM2.** We selected 6 rare variant single nucleotide polymorphisms (SNPs) on the coding region of the *TREM2* gene that have previously been associated with increased AD risk *rs2234256, rs2234255, rs142232675, rs143332484, rs75932628, rs2234253*[22,67–71].

Genetic data for ADNI and A4 were downloaded from The National Institute on Aging Genetics of Alzheimer's Disease Data Storage Site (NIAGADS DSS) and the Laboratory of Neuro Imaging Image and Data Archive (LONI IDA), respectively. ADNI genetic data was whole genome sequencing data from ADSP's August 15, 2022 release in the form of a VCF file. A4 data were Plink files, filtered for non-Hispanic white (NHW) individuals imputed on the TOPMed imputation server. BCFTOOLs were used to extract our 6 variants of interest. We interrogated the SNP data for the HABS-HD dataset and found only 2 of the 6 *TREM2* variants were sampled.

We collapsed across the 6 SNPs to form a binary *TREM2* risk variant carrier status[65], where the presence of any of the a-priori SNPs was labelled a 1 and the absence was labelled a 0. There were no weightings applied to the SNPs. Due to the limited *TREM2* SNPs data in the HABS-HD, we omitted this variable from our replication analysis. The number of *APOE-ε4* alleles for each participant was downloaded from LONI for

the A4 and ADNI samples. For the HABS-HD data *APOE-ε4* data was made available through the University of North Texas Institute for translational research. For all studies, individuals were classified as *APOE-ε4* non-carriers, heterozygotes, and homozygotes (Table 1).

## Statistics and reproducibility

**Path analysis under the amyloid cascade hypothesis.** Using Structural equation models (SEM) with interacting terms we built the possible impacts that genetic parent variables (sex, *APOE-ε4*, *TREM2*) have on different aspects of the AD pathological cascade, including age as a predictor to account for its potential confounding effect. We built the path following the canonical amyloid cascade hypothesis whereby Aβ is the initiating event, followed by increased levels of tau in the EC, then increased levels of tau in the neocortex (i.e. MetaTemp ROI) (Fig. 1). We assessed each stage of the AD cascade by building a series of SEMs modelling all main and 2-way interactions between genetic parent variables and upstream pathologies. To ensure a parsimonious description of the data we employed forward variable selection in each SEM based on Akaike Information Criterion (AIC). To allow for an accurate estimation of the causal effects when two continuous variables interact (i.e. Aβ and EC tau), we discretise Aβ into four bins (< 10, 10–40, 40–60, > 60) when including it as an interactive term with EC tau. Continuous Aβ is still included as a main effect variable when interactions are retained following model selection. A more detailed description of model selection is described in Supplementary Methods – Model Building. Following variable selection the following reduced models were investigated.

$$A\beta = \mu_0 + \alpha_0 \text{Age} + \beta_0 \text{Sex} + \gamma_0 \text{TREM2} + \delta_{01}\text{APOE4}_1 + \delta_{02}\text{APOE4}_2 + \epsilon_0 \quad (1)$$

$$\begin{aligned}\tau_{\text{EC}} = {}& \mu_1 + \theta_1 A\beta + \alpha_1 \text{Age} + \beta_1 \text{Sex} + \gamma_1 \text{TREM2} + \delta_{11}\text{APOE4}_1 + \delta_{12}\text{APOE4}_2 \\ & + \lambda_{11}A\beta \times \text{APOE4}_1 + \lambda_{12}A\beta \times \text{APOE4}_2 + \zeta_1 A\beta \times \text{Sex} + \phi_1 \text{Sex} \times \text{TREM2} + \epsilon_1\end{aligned} \quad (2)$$

$$\begin{aligned}\tau_{\text{Meta}} = {}& \mu_2 + \theta_2 A\beta + \kappa_2 \tau_{\text{EC}} + \alpha_2 \text{Age} + \beta_2 \text{Sex} + \gamma_2 \text{TREM2} + \delta_{21}\text{APOE4}_1 \\ & + \delta_{22}\text{APOE4}_2 + \nu_2 \mathbb{1}_{\{A\beta \geq 60\}} \times \tau_{\text{EC}} + \tau_2 \tau_{\text{EC}} \times \text{TREM2} + \pi_{21}\tau_{\text{EC}} \times \text{APOE4}_1 \\ & + \pi_{22}\tau_{\text{EC}} \times \text{APOE4}_2 + \omega_{21}\text{TREM2} \times \text{APOE4}_1 + \omega_{22}\text{TREM2} \times \text{APOE4}_2 \\ & + \psi_2 A\beta \times \text{TREM2} + \eta_2 \tau_{\text{EC}} \times \text{Sex} + \epsilon_2\end{aligned} \quad (3)$$

To assess potential multicollinearity introduced by interacting terms in our structural equation models, we calculated the scaled generalised variance inflation factor (GVIF) for all predictors, including categorical and interaction variables (see Supplementary Table 18). Most predictors showed scaled GVIF values below 5, well within conventional thresholds. A few variables exhibited scaled GVIF values near or below 7, which, while slightly higher, remained in an acceptable range for accurate estimation. These higher values primarily reflect the restricted range of EC tau but do not compromise the overall interpretability of the effect estimates.

To assess the statistical differences between estimated effects for different levels of genetic variables we ran models on 1000 multiplier bootstrapped samples. When genetic factors interact with upstream pathologies, we visualise these results by showing the marginal effects across different levels of upstream pathologies. In the R version 4.1.1 environment, we applied the SEMs to the combined dataset, excluding two outliers (RIDs 4414 and 4715 from the ADNI dataset) identified through diagnostic checks. Using the parameter estimates obtained from the SEMs, we calculated the marginal effects by subsequently varying the levels of genetic factors, Aβ (in EQ2), and EC tau (in EQ3). We then fit the HABS-HD data to the reduced models to assess if the estimates observed in the discovery data replicate in a new sample. Due to incomplete *TREM2* information in the HABS-HD we omitted this variable from replication models.

Finally, we calculated the direct, mediation, and total effects along the pathway from Aβ to EC tau to MetaTemp tau for different levels of

genetic variables, contrasting levels of downstream pathology (i.e. MetaTemp tau) for a given level of upstream pathology (i.e. Aβ) under varying genetic profiles. The mediation effect was quantified by incrementing Aβ by one unit in the EQ2, observing the resultant change in EC tau, and then applying this change to the EQ3 to measure the consequent variation on MetaTemp tau. The detailed derivations of the mediation effect are shown in Supplementary Methods - Mediation Effect Analysis. We also calculated the sample size weighted mean of the mediation effect for varying levels of *APOE4*, *TREM2*, and sex to elucidate the mediation effect with respect to individual factors.

## Reporting summary

Further information on research design is available in the Nature Portfolio Reporting Summary linked to this article.

## Data availability

ADNI, HABS-HD, and A4 studies data are available upon access approval through the following websites. ADNI website: https://adni.loni.usc.edu/. HABS-HD website: https://apps.unthsc.edu/itr/. A4 website: https://atri.usc.edu/study/a4-study/. Source data for the marginal effects shown in figures are provided with this paper. Source data are provided with this paper.

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

## Acknowledgements

Data collection and sharing for this project were funded by the Alzheimer's Disease Neuroimaging Initiative (ADNI) (National Institutes of Health Grant U01 AG024904) and DOD ADNI (Department of Defense award number W81XWH-12-2-0012). ADNI is funded by the National Institute on Aging, the National Institute of Biomedical Imaging and Bioengineering, and through generous contributions from the following: AbbVie, Alzheimer's Association; Alzheimer's Drug Discovery Foundation; Araclon Biotech; BioClinica, Inc.; Biogen; Bristol-Myers Squibb Company; CereSpir, Inc.; Cogstate; Eisai Inc.; Elan Pharmaceuticals, Inc.; Eli Lilly and Company; EuroImmun; F. Hoffmann-La Roche Ltd and its affiliated company Genentech, Inc.; Fujirebio; GE Healthcare; IXICO Ltd.; Janssen Alzheimer Immunotherapy Research & Development, LLC.; Johnson & Johnson Pharmaceutical Research & Development LLC.; Lumosity; Lundbeck; Merck & Co., Inc.; Meso Scale Diagnostics, LLC.; NeuroRx Research; Neurotrack Technologies; Novartis Pharmaceuticals Corporation; Pfizer Inc.; Piramal Imaging; Servier; Takeda Pharmaceutical Company; and Transition Therapeutics. The Canadian Institutes of Health Research is providing funds to support ADNI clinical sites in Canada. Private sector contributions are facilitated by the Foundation for the National Institutes of Health (www.fnih.org). The grantee organisation is the Northern California Institute for Research and Education, and the study is coordinated by the Alzheimer's Therapeutic Research Institute at the University of Southern California. ADNI data are disseminated by the Laboratory for Neuro Imaging at the University of Southern California. Research reported on this publication was supported by the National Institute on Aging of the National Institutes of Health under Award Numbers R01AG054073, R01AG058533, P41EB015922 and U19AG078109. The content is solely the responsibility of the authors and does not necessarily represent the official views of the National Institutes of Health. The A4 Study is a secondary prevention trial in preclinical Alzheimer's disease, aiming to slow cognitive decline associated with brain amyloid accumulation in clinically normal older individuals. The A4 Study is funded by a public-private-philanthropic partnership, including funding from the National Institutes of Health-National Institute on Aging, Eli Lilly and Company, Alzheimer's Association, Accelerating Medicines Partnership, GHR Foundation, an anonymous foundation and additional private donors, with in-kind support from Avid and Cogstate. The companion observational Longitudinal Evaluation of Amyloid Risk and Neurodegeneration (LEARN) Study is funded by the Alzheimer's Association and GHR Foundation. The A4 and LEARN Studies are led by Dr. Reisa Sperling at Brigham and Women's Hospital, Harvard Medical School and Dr. Paul Aisen at the Alzheimer's

Therapeutic Research Institute (ATRI), University of Southern California. The A4 and LEARN Studies are coordinated by ATRI at the University of Southern California, and the data are made available through the Laboratory for Neuro Imaging at the University of Southern California. The participants screening for the A4 Study provided permission to share their de-identified data in order to advance the quest to find a successful treatment for Alzheimer's disease. We would like to acknowledge the dedication of all the participants, the site personnel, and all of the partnership team members who continue to make the A4 and LEARN Studies possible. The complete A4 Study Team list is available on: www.actcinfo.org/a4-study-team-lists. This research was supported in part by the Intramural Research Program of the NIH, National Institute on Aging (NIA), National Institutes of Health, Department of Health and Human Services; project number ZIA AG000534, as well as the National Institute of Neurological Disorders and Stroke. *Funding:* J.G. is supported by the Alzheimer's Association (23AARF-1026883). W.J. is supported by the NIH (AG034570 and AG062542). JY is supported by the NIH ( R01AG062588, R01AG057234, P30AG062422, P01AG019724, U19AG079774; U54NS123985; 75N95022C00031); the Rainwater Charitable Foundation; the AFTD Susan Marcus Memorial Fund; the Larry L. Hillblom Foundation; the Bluefield Project to Cure Frontotemporal Dementia; the Alzheimer's Association; the Global Brain Health Institute; the French Foundation; the Mary Oakley Foundation; Alector, Transposon Therapeutics. C.J's participation in this project was part of a competitive contract awarded to DataTecnica by the National Institutes of Health to support open science research.

## Author contributions

J.G.: Conceptualisation, Investigation, Methodology, Writing. C.J.: Conceptualisation, Analysis, Methodology, Writing. Y.W.: Analysis, Methodology, Writing. J.Y.: Writing, Supervision. J.W.: Conceptualisation, Investigation, Methodology, Analysis, Writing, Supervision, W.J.: Conceptualisation, Investigation, Writing, Supervision.

## Competing interests

W.J. serves as a consultant to Biogen, Genentech, CuraSen, Bioclinica, and Novartis. J.S.Y. serves on the scientific advisory board for the Epstein Family Alzheimer's Research Collaboration. All other authors declare no competing financial interests.

## Additional information

## the Alzheimer's Disease Neuroimaging Initiative

**William J. Jagust** ⓘ [1,11]

