## [Transparent Peer Review file · Nature Communications]

Variable and interactive effects of Sex, APOE ϵ 4 and TREM2 on the deposition of tau in entorhinal and neocortical regions.

Corresponding Author: Dr Joseph Giorgio

Version 0:

Reviewer comments:

Reviewer #1

(Remarks to the Author)

The study by Joseph Giorgio and colleagues offers a significant contribution to understanding how genetic variations influence Alzheimer's disease (AD) progression, particularly through an analysis of over 1,300 participants. The results notably demonstrate that females and APOE- ϵ 4 homozygotes are more susceptible to amyloid beta (Ab) effects, leading to increased tau accumulation in the entorhinal cortex (EC). Furthermore, individuals carrying rare TREM2 risk variants or who are APOE- ϵ 4 homozygotes exhibit greater tau spread from the EC to the neocortex. These findings underscore the critical roles of sex, APOE, and microglia in AD and suggest important implications for personalized treatment strategies targeting Ab and tau. While the manuscript is well-written and accessible, several revisions are recommended to enhance the clarity and depth of the presentation:

(1) Given the complexity of the study and the breadth of analyses performed, it is crucial to provide clear guidance to the reader throughout the manuscript. Enhancing Figure 1 (upper panel) could be particularly beneficial. I recommend using this figure to illustrate the different analyses conducted across the manuscript, perhaps by employing distinct colors or line types to represent the various levels of analysis or factors of interest (e.g., TREM2, APOE, sex). Similarly, the causal path models in Figures 2 and 3 would benefit from further refinement to improve clarity. Additionally, I suggest including a graphical summary for the final section of the results, similar to Figure 1, but emphasizing the significant pathways identified.

(2) In the introduction, the odds ratio (OR) for TREM2 is not provided when discussing risk. Although the authors have consolidated multiple TREM2 risk variants into a single risk variant carrier status, it would be informative to include this data. Understanding whether the effect of TREM2 is greater or smaller than that of APOE4, and by what magnitude, could offer valuable insights for interpreting the results.

(3) To strengthen the manuscript, I recommend adding a section on hypotheses in the introduction. Clearly stated hypotheses would provide a solid foundation for the subsequent analysis and discussion.

(4) The use of Structural Equation Modeling with interacting terms requires careful attention. Interacting terms can introduce multicollinearity, which may inflate standard errors and complicate the interpretation of coefficients. The authors should conduct and report diagnostic tests for multicollinearity and describe how they mitigated its impact. Additionally, it is important to explicitly discuss how the model was identified and ensure that the parameters are uniquely estimable, given the added complexity.

(5) Minor Concerns:

- The definitions of terms such as "APOE," "PET," "ROI," "SNPs," and "CL" (among others) are either missing or not provided in the correct order within the manuscript, especially considering that the Methods section follows the Results.
- Please replace "TREM2 (Triggering Receptor Expressed on Myeloid Cells 2)" with "Triggering Receptor Expressed on Myeloid Cells (TREM2)."
- The manuscript should specify which part of the figure is being referenced (e.g., Figure 3a, 3b).
- Update the title section from "FTP (Flortaucipir PET) Tau" to "Flortaucipir (FTP) PET Tau."
- Consider adding a note to clarify that "sex(F)" refers to females in the results.
- In the discussion, maintain consistency with the rest of the manuscript by discussing APOE results before TREM2-related findings.

- The final interpretation of sex differences in the results section is unclear. Please review and clarify this part of the text.
- Ensure that all abbreviations are fully explained in the Figures and Tables.
- The conclusion paragraph should be revised. The authors mention insights into sex, APOE, and microglia in AD progression but do not reference sex differences in the previous sentences of this section.
- The meaning of the arrow from Ab to Tau meta in Figure 3e is unclear and should be clarified.
- Figure 3 lacks a plot showing significant interactions between EC tau and high A β (>60CL).

Reviewer #2

(Remarks to the Author)

Georgio & Jonson et al. apply multivariate directed models to explain purported causal relationships between Alzheimer's disease pathology (A-beta and tau, measured with PET) and genetic influences on AD (APOE, TREM2 and sex). The authors fit three different multivariate models, each revealing interesting interactions between variables, many of which have been reported and some that are novel. Some of the main findings include:

- * The influence of amyloid on neocortical tau is driven by amyloid effect on entorhinal tau
- * APOEe4 homozygotes accumulate more tau than for a given level of amyloid
- * Possible interactions between TREM2 and sex, and between TREM2 in APOE, in driving increased tau accumulation.

This study has many strengths, greatest of which is the existence of a large and ethnically diverse validation sample supporting many (but not all) of the findings. The use of multivariate directed models provides a decidedly more comprehensive understanding of relationships between variables commonly studied in isolation or without directed modeling. TREM2 mutations have also been challenging to study due to being fairly rare in the population, and their inclusion presently is quite interesting. Finally, the study provides interesting findings potentially relevant to current treatments for AD. However, many of the more interesting effects with TREM2 are not replicated, and some aspects of model construction and variable choice are not well explained. In addition, while many results make sense and support previous work, others are harder to interpret in the context of previous consensus research. These issues are not sufficiently addressed in the current manuscript, and are outlined in detail below in no particular order:

1) The theme, direction and narrative of the paper is not very clear. The choice of variables seems rather arbitrary. One theme involves sex and APOE in response to new immunotherapies, but then it's hard to see where TREM2 fits in? TREM2 and E4 homozygotes are linked in their strong impact on genetic risk for AD, but then sex doesn't quite fit in? Later the authors talk about how all of these are genetic influences, which is true, but then why not be more comprehensive and include an AD PRS? Given that the small number of variables are handpicked by the authors, I would have expected a stronger hypothetical relationship between them clearly outlined in the introduction, but I am missing this.

2) Neuropathology studies suggest that TREM2 is more likely to result in an atypical AD presentation with more neocortical than medial temporal tau (10.1007/s00401-022-02495-4). In this study, a main effect of TREM2 was seen in neocortical tau, but not entorhinal tau. Since these cortical-predominant cases may defy typical Braak-like progressions, it might be worthwhile to see whether a model where neocortical tau leads to entorhinal tau (rather than vice versa) is worth exploring, especially for TREM2. Does such a model fit the data as well or better in TREM2 models? What about a model with bidirectional relationships between these two and simultaneous relationship with A β ? These are plausible scenarios in cortical predominant cases.

3) Much of what was presented in the study is already well known or has been published. For instance, the idea that APOE leads to increased tau per level of Ab has been recently published (10.1001/jamaneurol.2023.4038). To this reviewer, the two most novel and innovative findings in the study are the findings related to the rare TREM2 mutations, and the integration of different variables into a large multi-variate directed model. However, both of these new and exciting findings have different flaws:

a) The TREM2 effects were all quite interesting, but were also the flimsiest findings in terms of support from the data. The samples were quite small (how large were the different sex groups of TREM2+ individuals?, or the E4 homozygote + TREM2 groups?), some of the effects were quite close to the alpha value for significance (e.g. the sex-TREM2 interaction on entorhinal tau) and, most importantly, there was no replication. As the least well understood element of the paper, this one is the most important to validate. I did not fully understand why these models were not attempted validation in HABS-HD. It seemed that there were TREM2 issues in all datasets, but they were determined too much to overcome in HABS-HD. It would be nice to see if the effects are similar even among subjects who did have TREM2 mutations in HABS-HD.

b) The final model in the paper was very interesting and produced novel results. The novelty comes from the approach of a multivariate directed framework. In that regard, what is the purpose of interpreting the first two models, which are smaller and represent a less complete overall picture of the data? If results change after adding more variables and pathways, shouldn't this suggest that the earlier results were driven by these other explanatory data not being included?

3) Some of the APOE effects are puzzling. It seems that many of the effects described are not present in APOE e4 heterozygotes. For example, in Fig 3C, E4 heterozygotes are indistinguishable from controls. This is interesting because it is also more similar to many of the animal models that are described to support the APOE effect on tau, which tend to use homozygous expression, often on top of an AD genetic background. However, the APOE findings are presented in the discussion more broadly. Could the authors perhaps speculate as to why the effects do not appear to be dose dependent in their study, and how the reader should interpret this in the broader context of the APOE - tau relationship?

One must be careful about the hidden effect of APOE on disease progression. APOE carriers in the dataset are more likely to have amyloid, and to have MCI and AD, and these cases are all more likely to have more tau. The authors present an attempt at a causal model, but other factors could explain the findings (e.g. non-tau-mediated cognitive effects). The direct effect of APOE on tau would be made more solid by showing this trend is present just in Ab+ controls, and/or just in Ab+ MCI. Especially since these effects seem to contradict some other work out there suggesting amyloid mediates E4 effects on tau (e.g. 0.1007/s00259-021-05192-8).

4) Causality is a polarizing term in this context. The authors are certainly using causal models, and there is certainly a hypothesis of implied causality between variables in this dataset. But to say that the authors are reporting “causal effects” is probably overblown. We don’t know if the effects are causal, even if this causal model supports the claim that they are. I would recommend removing phrases and section headings that purport causal effects, or at least amend them to say “purported causal effects” or something like that. In addition, it is not entirely clear how the models are chosen, and whether alternative models better support the data.

5) The replication sample is a huge strength of the study and many of the replications are impressive. I am curious of the authors’ perspective why, for the model described in Tables S1 and S2, the APOE effect size is cut in half in the HABS-HD dataset compared to the discovery dataset.

6) Figure 1 is a nice description of all of the relationships in this study. Still, it would be useful for each figure thereafter, a similar diagram is given highlighting the relationships involved. I thought that was there but those figures do not depict any interactions/mediations that are clearly involved in those models, so its a bit hard to understand how they relate.

7) No data is shown and it would be nice to at least chart confidence intervals in the Figures showing marginal relationships. Figure 2C is even a barplot. I’m pretty sure Nat Comms doesn’t even allow barplots, and for good reason.

8) While certainly interesting, I have trouble interpreting the effect of female sex on entorhinal but not cortical tau in the full multivariate model toward the end of the paper. The author’s seem to interpret this as vulnerability, but could this not also be interpreted as resistance? In other words (if we believe the models here), that women are able to retain more pathology in the entorhinal cortex before it spreads out into neocortex?

9) Can the authors explain the binning windows of centiloids? Much has been made about logical cut-offs for centiloids. With reference to this work, wouldn’t it make more sense for the bins to be 0-20, 20-40, 40-60, >60? Do the amyloid interactions change at all using this binning strategy?

10) Conceptually I have some trouble fully understanding the proposed causality of TREM2 in the models. Genetic markers are obviously upstream of measured phenotypes. With APOE, there is strong evidence supporting the idea that its effects are causally driving amyloid accumulation. However, most data and hypotheses purport microglia effects to become relevant *after* Ab deposition occurs. If this is the case, and if the authors agree, would it not make more sense to generate some sort of latent TREM2 phenotype (e.g. microglial response) that is influenced by (e.g. downstream of) TREM2 but also downstream of Abeta? If such a factor, or TREM2 itself, is placed after Ab, does it the model fit the data similarly or better?

Reviewer #3

(Remarks to the Author)

In this manuscript, Giorgio et al. investigated how genetic variation (here reflected by sex, APOE e4 dosage, and TREM2 risk variant carriership) moderated different aspects of the amyloid cascade hypothesis (here reflected by amyloid deposition leading to entorhinal tau leading to neocortical tau). The manuscript is well written and overall clear, and the use of a racially diverse replication cohort strengthens their findings. However, there are also some limitations to this work among which the lack of longitudinal data and the pooling of participants of different cognitive stages. In addition, the final part of the results section was relatively difficult to grasp.

I have the following questions, comments and suggestions:

1. The cohort includes a mixture of CU, MCI and dementia participants and the authors do not control for this in their analyses. I think it would be of importance to show that the observed associations are not driven by differences in disease stage.
2. One of the limitations of this study is its cross-sectional nature (while the AD/amyloid cascade is a long longitudinal process). Previous studies have been published performing comparable cross-sectional models, e.g. looking at effects of sex*amyloid on tau (Buckley et al. 2019 JAMA Neurology), or APOE*amyloid on tau (Therriault et al. 2020 Molecular Psychiatry). One of the strengths of the current study is that sex, APOE and TREM2 are modelled simultaneously thereby investigating their independent effects. However, if the authors additionally have longitudinal tau-PET data at hands (which at least should be available in ADNI), it could be of interest to include that here in order to further understand the influence of genetic variation on the longitudinal progression of tau.
3. The final part of the results section (“Variable genetic effects at different stages in the canonical amyloid cascade pathway”) is difficult to grasp. Very few statistics/results are reported in the text, and there is no accompanying visualization that can guide the reader. It would be helpful if the authors can add a figure visualizing all pathways and if the authors can include the strengths/significances of all pathways (all direct effects, mediation effects and total effects – for all genetic groups) (also see point #6).
4. Regarding the final part of the results: is it correct that TE reflects DE + ME? If so, I would specify that as this is currently

not entirely clear.

5. Regarding the final part of the results: I found this sentence complex "When we contrast these effects amongst [...] (2-alleles) vs (1-allele)=0.0029,p=0.0335)." Please add the comparison group in the sentence (i.e., between APOE-ε4 homozygotes and ...?), and there seems to be a bracket missing in that sentence. In addition, what do the 0.003 and 0.0029 values precisely reflect? From the text it seems to reflect the difference in the total effect, but from Supplementary Table 10 it seems to reflect the difference in the mediation effect. Also, does this reflect a difference in a standardized beta or an unstandardized beta?

6. Regarding the final part of the results: in the text it is stated there are 'no significant differences in the TE across the other genetic groups', but in Supplementary Table 10 there does seem to be an effect for TREM2.

7. Regarding the causal path models displayed in Figures 2-3: it would be helpful to include the path coefficients (numbers along the arrows representing the strengths and significances of the relationships) in order to visualize which paths are significant and which are not.

8. Regarding the causal path models displayed in Figures 1-2-3: while stated in the legend, I am not sure whether these models are currently visualizing any interaction effects. I think the causal path models are currently only visualizing main effects of APOE/sex/TREM2 on amyloid/tau. To also visualize interaction effects within these causal path models, I think one would also need to add arrows going from e.g. APOE towards the arrow going from AB>TAU (in case one would want to visualize the interaction effect of APOE*AB on tau).

9. Regarding Figures 2-3: please add confidence intervals to the plots.

Version 1:

Reviewer comments:

Reviewer #1

(Remarks to the Author)

Thank you for your revised submission. I appreciate the thoughtful and thorough manner in which you addressed all the comments and suggestions provided in the initial review. The updated figures, especially the refinements to Figures 1 through 3 and the addition of a graphical summary in the final section, significantly enhance the clarity of the manuscript. Your inclusion of the odds ratio for TREM2, the clarified modeling approach, and the expanded discussion of hypotheses provide a stronger conceptual foundation and more robust interpretation of the findings.

Minor revisions related to terminology, figure referencing, and consistency throughout the manuscript have also been successfully implemented. These changes contribute to a clearer and more coherent presentation of your important work.

Overall, the manuscript is now greatly improved.

Reviewer #2

(Remarks to the Author)

The manuscript, already good, has been improved based on the revisions. It would be nice for the authors to add comments from my previous point 10 into the discussion, as I think it is quite interesting and further supports the utility of the authors' model. But I do not think this is something that should hold up the publication process. The authors can make the decision on what they wish to do here.

Reviewer #3

(Remarks to the Author)

The authors have addressed all my comments.

Response to reviewers.

We would like to thank the three reviewers for their in depth and insightful review of our manuscript. We have amended the manuscript accordingly and believe that this has led to an improved body of work.

Reviewer #1

The study by Joseph Giorgio and colleagues offers a significant contribution to understanding how genetic variations influence Alzheimer's disease (AD) progression, particularly through an analysis of over 1,300 participants. The results notably demonstrate that females and APOE-ε4 homozygotes are more susceptible to amyloid beta (Ab) effects, leading to increased tau accumulation in the entorhinal cortex (EC). Furthermore, individuals carrying rare TREM2 risk variants or who are APOE-ε4 homozygotes exhibit greater tau spread from the EC to the neocortex. These findings underscore the critical roles of sex, APOE, and microglia in AD and suggest important implications for personalized treatment strategies targeting Ab and tau. While the manuscript is well-written and accessible, several revisions are recommended to enhance the clarity and depth of the presentation:

(1) Given the complexity of the study and the breadth of analyses performed, it is crucial to provide clear guidance to the reader throughout the manuscript. Enhancing Figure 1 (upper panel) could be particularly beneficial. I recommend using this figure to illustrate the different analyses conducted across the manuscript, perhaps by employing distinct colors or line types to represent the various levels of analysis or factors of interest (e.g., TREM2, APOE, sex). Similarly, the causal path models in Figures 2 and 3 would benefit from further refinement to improve clarity. Additionally, I suggest including a graphical summary for the final section of the results, similar to Figure 1, but emphasizing the significant pathways identified.

We have now updated all figures to improve the clarity of the causal path models. In addition we have also generated a summary figure indicating the significant pathways identified.

Figure 1 Canonical Alzheimer's disease cascade. The left panel indicates the stages and spatial distribution of Alzheimer's Disease (AD) pathology throughout the cascade. We modelled the initial stages of tau pathology using the entorhinal cortex (Tau_{EC}) region of interest. Neocortical

tau was modelled using the meta temporal (Tau_{Meta}) region of interest. The right panel shows the directed acyclic graph used to model the potential pathways between genetic variables and pathology. Each line originates at a predictor variable with the yellow nodes indicating an AD pathology. Solid lines indicate the pathway from an upstream variable to a downstream pathology, dashed lines indicate an interactive effect between upstream variables on a downstream pathology, with the arrow indicating a modulation of the pathway showed by the solid line. Solid and dashed lines initiating from the black genetic variables box indicate a direct or interactive effect respectively of genetic variable on the outcome variable. The black box indicates the three genetic variables that are modelled as direct or interactive predictors of AD pathologies, these are the number of Apolipoprotein E (*APOE*)- $\epsilon 4$ alleles (0,1,2), Sex (female, male), and Triggering Receptor Expressed on Myeloid Cells 2 (*TREM2*) risk variant carrier status (0,1).

Figure 2 Genetic influences on entorhinal cortex tau. Lines show the estimated marginal levels of tau pathology for individuals with varying genetic profiles at different levels of amyloid beta ($A\beta$). Whisker plot shows estimated levels of entorhinal cortex tau (Tau_{EC}) pathology based on sex and Triggering Receptor Expressed on Myeloid Cells 2 (*TREM2*) risk variant carrier status. Different effects in the discovery sample for **a.** Apolipoprotein E (*APOE*)- $\epsilon 4$, **b.** sex, **c.** interaction of sex and *TREM2* risk variant carrier status. Different effects in the replication sample for **d.** *APOE*- $\epsilon 4$ and **e.** sex. **f.** Causal path model to estimate Tau_{EC} . Solid lines indicate a direct effect of an upstream variable on a downstream pathology, dashed lines indicate an interactive effect between upstream variables (i.e. $A\beta$ centiloid or genetic variable) on Tau_{EC} . Lines initiating from the black genetic variables box may indicate a direct or interactive effect of genetic variables on

the outcome variable. Values in the parentheses indicate level of genetic variable (*APOE-ε4* alleles (0,1,2) and *TREM2* risk variant carrier status (0,1)).

Figure 3 Genetic influences on meta temporal tau. Lines show the estimated marginal levels of meta temporal tau (Tau_{Meta}) pathology for individuals with varying genetic profiles at different levels of entorhinal cortex tau (Tau_{EC}). Different effects in the discovery sample for **a.** Apolipoprotein E (*APOE*)- $\epsilon 4$, **b.** Triggering Receptor Expressed on Myeloid Cells 2 (*TREM2*), and **c.** their interaction. Different effects in the replication sample for **d.** *APOE*- $\epsilon 4$. **e.** Causal path model to estimate Tau_{Meta} . Solid lines indicate a direct effect of an upstream variable on a downstream pathology, dashed lines indicate an interactive effect between upstream variables (i.e. amyloid beta ($A\beta$), Tau_{EC} , or genetic variable) on meta temporal tau. Lines initiating from the black genetic variables box may indicate a direct or interactive effect of genetic variables on the outcome variable. Values in the parentheses indicate level of genetic variable (*APOE*- $\epsilon 4$ alleles (0,1,2) and *TREM2* risk variant carrier status (0,1)).

Figure 5 Genetic influences on the AD cascade. Causal path models indicating significant pathways in which genetic variation leads to increased Alzheimer's Disease (AD) pathology. Each line originates at a genetic variable with the yellow nodes indicating an outcome variable. Solid lines indicate the pathway from an upstream variable to a downstream pathology, dashed lines indicate an interactive effect between upstream variables on a downstream pathology with the arrow indicating a modulation of the pathway showed by the solid line. a. effect of Apolipoprotein E (*APOE*)- $\epsilon 4$ homozygosity on AD pathology, b. effect of Triggering Receptor Expressed on Myeloid Cells 2 (*TREM2*) risk variant carrier status on AD pathology, c. effect of female sex on AD pathology, d. interactive effect of *APOE*- $\epsilon 4$ and *TREM2* risk variant carrier status on AD pathology, and e. effect of female sex and *TREM2* risk variant carrier status on AD pathology.

(2) In the introduction, the odds ratio (OR) for TREM2 is not provided when discussing risk. Although the authors have consolidated multiple TREM2 risk variants into a single risk variant carrier status, it would be informative to include this data. Understanding whether the effect of TREM2 is greater or smaller than that of APOE4, and by what magnitude, could offer valuable insights for interpreting the results.

We have now updated the introduction to include the odds ratio of having clinical AD based on TREM2 risk variants.

In particular the introduction now writes pg. 4

Multiple *TREM2* polymorphisms have been linked to an increased risk of AD with varying effect ranging from 1.2 OR - 3 OR, of which R47H with a 2.71 OR for clinical AD is the most widely studied^{1,2}.

(3) To strengthen the manuscript, I recommend adding a section on hypotheses in the introduction. Clearly stated hypotheses would provide a solid foundation for the subsequent analysis and discussion.

We have now updated the introduction to include a section of hypotheses.

In particular the introduction now writes pg. 5

Here, we use causal path modelling to assess how genetic variation impacts the AD pathological cascade (**Figure 1**). Using data from within subject multimodal PET and whole genome sequencing (WGS) in a sample of 1354 individuals we probe different stages of the AD cascade to understand how genetic variation in sex, *APOE-ε4* and *TREM2* exacerbate AD pathology. We tested the effect of genetic variation through pathways mediated by Aβ or via non-Aβ pathways, that is, primary tau accumulation in the EC and spread into the neocortex after accounting for Aβ. Our causal path is structured assuming that there are stages in the AD cascade, with the initial event being Aβ deposition, followed by Aβ-related deposition of tau in the EC, followed by tau spreading from the EC into the neocortex (**Figure 1**). We hypothesise that each genetic variable will enter the AD cascade through varied routes, interacting with either Aβ to increase levels of EC tau or with EC tau to increase levels of neocortical tau. Specifically, we hypothesise that sex will interact with Aβ to increase levels of primary tau and that the number of *APOE-ε4* alleles will have an Aβ independent effect on tau. Through more exploratory analyses, we hypothesise there will be variable and interactive effects of *TREM2* and *APOE-ε4* that increase tau spreading from primary regions into the neocortex (i.e. greater neocortical tau for a given level of primary tau).

(4) The use of Structural Equation Modeling with interacting terms requires careful attention. Interacting terms can introduce multicollinearity, which may inflate standard errors and complicate the interpretation of coefficients. The authors should conduct and report diagnostic tests for multicollinearity and describe how they mitigated its impact. Additionally, it is important to explicitly discuss how the model was identified and ensure that the parameters are

uniquely estimable, given the added complexity.

To assess multicollinearity in our structural equation models we have calculated the scaled generalized variance inflation factor for all predictors shown in supplementary table 18.

Supplementary Table 18: Scaled GVIF for Structural Equation Models in the discovery sample and replication sample.

	GVIF ^{1/(2Df)} discovery	GVIF ^{1/(2Df)} replication
Aβ as outcome		
Age	1.04	1.01
Sex	1.02	1.01
TREM2	1.00	
APOE- ϵ 4	1.01	1.00
EC tau as outcome		
A β	1.73	1.89
Age	1.06	1.08
Sex	1.50	1.12
TREM2	1.69	
APOE- ϵ 4	1.75	1.28
A β \times APOE- ϵ 4	1.96	1.45
A β \times Sex	1.75	1.71
Sex \times TREM2	1.72	
Meta Temp tau as outcome		
A β	1.77	1.89
EC tau	1.88	1.96
Age	1.06	1.08
Sex	6.19	6.11
TREM2	6.43	
APOE- ϵ 4	5.63	4.80
A β _{\geq60} \times EC tau	1.78	1.89
EC tau \times APOE- ϵ 4	5.76	4.88
EC tau \times TREM2	6.74	
TREM2 \times APOE- ϵ 4	1.39	
A β \times TREM2	1.91	
EC tau \times Sex	6.39	6.25

Supplementary Table 18. Scaled generalized variance inflation factor (GVIF) values for all covariates in structural equations across both discovery and replication datasets.

1. **Multicollinearity Check Using Scaled GVIF:**

We assessed multicollinearity using the scaled Generalized Variance Inflation Factor $GVIF^{1/(2Df)}$, as displayed in Supplementary Table 18. This approach allows us to account for the degrees of freedom in categorical and interaction terms, making the results comparable to traditional VIF values. We believe this is the most appropriate method given the complexity of our model, which includes multiple genetic factors and their interactions.

2. **VIF Interpretation:**

Most variables in our models show scaled GVIF values below 5, which is widely accepted as a reasonable threshold to rule out severe multicollinearity. However, a few variables exhibit scaled GVIF values slightly below 7, which, although slightly higher, are still within an acceptable range and do not present significant concerns for the accuracy or interpretability of our estimates. Specifically, the slightly higher GVIF values are attributed to the narrow range of EC tau, which naturally introduces some numerical challenges without compromising the overall model.

3. **Interpretation of Interaction Terms:**

While we report estimated coefficients and standard errors for both the genetic factors and their interactions with Aβ or EC tau in the Structural Equation Modeling results, these terms are not interpreted as independent factors in our subsequent analyses. For example, as illustrated in Figure 3, our primary focus is on the effect of Aβ on Meta Temp tau across different genetic groups. In this context, we are less concerned with interpreting the coefficients of individual interaction terms but are instead examining the overall patterns of change in Meta Temp tau as Aβ increases across genetic subgroups. Similarly, in Supplementary Table 16,17, where we assess the mediation effects of genetic factors, Aβ or EC tau are integrated into the overall mediation calculation, further reducing the need to interpret individual interaction coefficients.

In particular the methods now writes pg 23

To assess potential multicollinearity introduced by interacting terms in our structural equation models, we calculated the scaled generalized variance inflation factor (GVIF) for all predictors, including categorical and interaction variables (see **Supplementary Table 18**). Most predictors showed scaled GVIF values below 5, well within conventional thresholds. A few variables exhibited scaled GVIF values near or below 7, which, while slightly higher, remained in an acceptable range for accurate estimation. These higher values primarily reflect the restricted range of EC tau, but will not compromise the overall interpretability of the effect estimates.

(5) Minor Concerns:

- The definitions of terms such as “APOE,” “PET,” “ROI,” “SNPs,” and “CL” (among others) are either missing or not provided in the correct order within the manuscript, especially considering that the Methods section follows the Results.

We have now clarified abbreviations introducing the full names on first presentation.

- Please replace “TREM2 (Triggering Receptor Expressed on Myeloid Cells 2)” with “Triggering Receptor Expressed on Myeloid Cells (TREM2).”

We have now clarified this abbreviation.

- The manuscript should specify which part of the figure is being referenced (e.g., Figure 3a, 3b).

We have now amended the results calling the specific figure panel being referenced.

- Update the title section from “FTP (Flortaucipir PET) Tau” to “Flortaucipir (FTP) PET Tau.”

We have now amended the section heading as suggested.

- Consider adding a note to clarify that “sex(F)” refers to females in the results.

We have now amended the notation to read sex(Female).

- In the discussion, maintain consistency with the rest of the manuscript by discussing APOE results before TREM2-related findings.

We have now amended the order of results discussed in this section as suggested.

- The final interpretation of sex differences in the results section is unclear. Please review and clarify this part of the text.

We have now re-written the final section ‘**Variable genetic effects at different stages in the canonical amyloid cascade pathway**’.

- Ensure that all abbreviations are fully explained in the Figures and Tables.

We have now described all abbreviations in the figures and tables

- The conclusion paragraph should be revised. The authors mention insights into sex, APOE, and microglia in AD progression but do not reference sex differences in the previous sentences of this section.

We now reference sex differences in primary tau accumulation in the conclusion paragraph.

In particular the conclusion writes pg19

We show in a diverse sample of over 1300 participants that females and *APOE*- ϵ 4 homozygotes are more susceptible to the primary accumulation of tau, with greater EC tau for a given level of $A\beta$.

- The meaning of the arrow from Ab to Tau meta in Figure 3e is unclear and should be clarified.

We have now clarified this in the description of Figure 3. This arrow indicates a main effect of $A\beta$ on MetaTemp tau.

- *Figure 3 lacks a plot showing significant interactions between EC tau and high $A\beta$ (>60CL).*

In order to generate similar interaction plots as shown in Figure 3 (e.g. 3b showing the interaction between *TREM2* and EC tau on MetaTemp tau) we would need to integrate effects over both direct and mediation pathways, making a similar calculation infeasible. Rather we present below (**Response to Reviewer 1, Figure 1.**) the results when fixing $A\beta = 80$ (as compared to $A\beta = 40$ shown in figure 3). As the interaction effect between EC tau and high $A\beta$ is small ($\beta=0.019$ $p=0.049$), these new plots do not show any substantial differences to Figure 3. This indicates that the interactions between genetic variation and EC tau are robust across different levels of $A\beta$.

In particular the results writes pg9

In addition, we observed significant interactions between EC tau and *APOE-ε4*(2-alleles) ($\beta=0.364$, $P<0.001$), EC tau and *TREM2*(1) ($\beta=0.309$, $P<0.001$), and EC tau and high $A\beta$ (>60CL) ($\beta=0.019$ $p=0.049$) although the latter is a small effect. Individuals who are *APOE-ε4* homozygotes and individuals with a *TREM2* risk variant carrier status, and to a lesser extent, individuals with high levels of $A\beta$ had greater downstream MetaTemp tau burden for a given level of EC tau (**Figure 3 a,b,, Supplementary Figure 2**).

Response to Reviewer 1, Figure 1. Marginal effects with $A\beta = 80$.

MetaTemp Tau vs EC Tau for TREM2-APOE- $\epsilon 4$ Combinations in the discovery sample

MetaTemp Tau vs EC Tau for Different APOE- $\epsilon 4$ Genotypes in the discovery sample

MetaTemp Tau vs EC Tau for Different APOE- $\epsilon 4$ Genotypes in the replication sample

MetaTemp Tau vs EC Tau for Different TREM2 in the discovery sample

Reviewer #2

Georgio & Jonson et al. apply multivariate directed models to explain purported causal relationships between Alzheimer's disease pathology (A-beta and tau, measured with PET) and genetic influences on AD (APOE, TREM2 and sex). The authors fit three different multivariate models, each revealing interesting interactions between variables, many of which have been reported and some that are novel. Some of the main findings include:

- * The influence of amyloid on neocortical tau is driven by amyloid effect on entorhinal tau*
- * APOEε4 homozygotes accumulate more tau than for a given level of amyloid*
- * Possible interactions between TREM2 and sex, and between TREM2 in APOE, in driving increased tau accumulation.*

This study has many strengths, greatest of which is the existence of a large and ethnically diverse validation sample supporting many (but not all) of the findings. The use of multivariate directed models provides a decidedly more comprehensive understanding of relationships between variables commonly studied in isolation or without directed modeling. TREM2 mutations have also been challenging to study due to being fairly rare in the population, and their inclusion presently is quite interesting. Finally, the study provides interesting findings potentially relevant to current treatments for AD. However, many of the more interesting effects with TREM2 are not replicated, and some aspects of model construction and variable choice are not well explained. In addition, while many results make sense and support previous work, others are harder to interpret in the context of previous consensus research. These issues are not sufficiently addressed in the current manuscript, and are outlined in detail below in no particular order:

1) The theme, direction and narrative of the paper is not very clear. The choice of variables seems rather arbitrary. One theme involves sex and APOE in response to new immunotherapies, but then its hard to see where TREM2 fits in? TREM2 and E4 homozygotes are linked in their strong impact on genetic risk for AD, but then sex doesn't quite fit in? Later the authors talk about how all of these are genetic influences, which is true, but then why not be more comprehensive and include an AD PRS? Given that the small number of variables are handpicked by the authors, I would have expected a stronger hypothetical relationship between them clearly outlined in the introduction, but I am missing this.

The aim of this work is to use genetic variation to provide a deeper understanding of AD pathophysiology. Using genetic instrumental variables that imply specific biological processes we are aiming to determine how the functions of APOE, TREM2 and sex interact with AD pathologies and each other to gain further insights in the mechanisms driving the AD cascade. It is the aim of being able to infer biological processes from genetic variables that make an AD PRS undesirable in this context as these are composite scores derived to capture multiple biological processes that increase risk of AD. Our selection of variables was due to the biological processes implicated through each variable and prior work describing interactions between these processes in the presence of AD pathology. We contextualise this work considering recent clinical trials. We have now amended the introduction to clarify the motivation of this work.

In particular the introduction now writes pg2:

Alzheimer's Disease (AD) follows a canonical cascade, with the accumulation of A β being the primary event accelerating tau accumulation and spread from the entorhinal cortex (EC) into the neocortex leading to cognitive decline³. Substantial evidence supports the role of pathological A β aggregates as a primary event in both sporadic and genetically dominant forms of AD⁴. In particular, individuals with dominantly inherited AD mutations show a predictable and sequential evolution of AD pathophysiology with A β accumulation leading to tau accumulation and subsequent spread⁵. However, these genetically dominant forms of AD are rare and the evolution of sporadic late-onset AD is influenced by a multitude of genetic and lifestyle factors, with the relative penetrance of the canonical AD cascade profoundly affected by genetic variation and comorbidities^{6,7}. Studying genetic variations that infer discrete biological processes provides key insight into how physiological processes such as lipid processing, microglial activation, and sex influence the evolution of AD pathology. Further, to fully appreciate the complex and interacting processes that are involved in the evolution of AD pathology, genetic variations must be modelled together and in combination with AD biomarkers (i.e. A β and tau PET).

The $\epsilon 4$ allele of *APOE* is the strongest risk factor for clinical sporadic late-onset AD, conferring an increased risk (i.e. odds ratio OR) of AD compared to *APOE*- $\epsilon 3$ homozygotes of up to 3.46 OR for $\epsilon 4$ heterozygotes and 13.04 OR for *APOE*- $\epsilon 4$ homozygotes⁸, with *APOE*- $\epsilon 4$ homozygosity having a near complete penetrance for A β positivity⁹. Interestingly, the *APOE*- $\epsilon 4$ allele also has

a marked increase in the risk of developing AD in A β positive individuals, suggesting additional effects beyond A β ¹⁰. Multiple lines of evidence link the APOE4 isoform to reduced homeostatic clearing of A β ^{6,11-13}. This aberrant function of APOE4 occurs through dysfunction in multiple cell types in the brain that play diverse roles in AD pathogenesis, including neurons, microglia, and astrocytes ^{11,14}. Regarding A β , APOE4 increases the accumulation of cortical A β by reducing the dissolution of soluble A β ¹⁵ and impairs the clearance of A β by disrupting the blood brain barrier ¹⁴. Further, APOE4 has been implicated in increased tau accumulation, with increased levels of APOE4 resulting in increased tau phosphorylation and interneuronal spread of tau ^{11,14,16,17}. Therefore, APOE is not only involved in amyloidosis but also in downstream and parallel events such as the development of tau pathology.

Alongside the *APOE- ϵ 4* allele, rare genetic polymorphisms in *TREM2* (Triggering Receptor Expressed on Myeloid Cells 2) have also been shown to be significant risk factors for sporadic late-onset AD ¹. *TREM2* is a transmembrane protein expressed in microglia and performs critical functions in the immune response to AD pathology. *TREM2* is involved in signaling cascades as well as the transition of microglia to a disease activated state, with a lack of functional *TREM2* profoundly impacting microglia function ¹⁸⁻²⁰. Multiple *TREM2* polymorphisms have been linked to an increased risk of AD with varying effect ranging from 1.2 - 3 OR, of which R47H with a 2.71 OR for clinical AD is the most widely studied^{1,2}. Previous work investigating this variant in animal models shows that rare polymorphisms lead to a hypofunctional form of *TREM2* promoting tau seeding and spreading ²¹. Multiple other genetic variants on the *TREM2* gene have been linked to increased risk of AD and negatively affect the function of *TREM2* in-vitro ^{1,19,20,22,23}. These variants may have an additive loss of function or alternatively may be in linkage disequilibrium

with the same functional variant ²⁴. How trait variation in *TREM2* impacts different phases of the AD pathological cascade is yet to be fully elucidated in humans and offers an approach to study how dysfunctional *TREM2* impacts microglial functions leading to increased burden of A β and tau. Furthermore, it remains to be seen how *TREM2* and *APOE4* interact with each other and A β or tau leading to a greater burden of AD pathology.

Sex plays an important role in the pathogenesis of AD, with dementia incidence higher in females in late life ²⁵. Furthermore, females have consistently been shown to have higher tau tangle load at autopsy than men ^{26–28}. This post mortem work is well supported in-vivo with multimodal neuroimaging studies showing females are susceptible to higher levels and faster accumulation rates of tau for a given level of A β than their male counterparts ^{29–32}. Whether these increases are specific to A β influences on primary accumulation of tau in the EC, or tau spreading mechanisms from the EC into the neocortex is not well resolved. However, it is not clear if this finding in females is due to, or, exacerbated by varied immune responses such as *TREM2*-related microglial dysfunction, with interactions between sex and immune processes previously reported ³³. Finally, it is not well resolved how sex and the *APOE- ϵ 4* allele interact to increase AD risk with effects of age and dosage impacting this relationship ⁸, coupled with inconsistent in-vivo imaging findings ^{30,32} further investigation is warranted.

Beyond a more complete appreciation of the biology governing the accumulation of AD pathology, understanding how genetic variation relates to variable development of AD pathologies is of pressing clinical need. The recent successful trials of Lecanemab and Donanemab have shown that reducing cortical amyloid-beta plaques (A β) led to significant slowing of cognitive decline;

however, our grasp on the precise scope and conditions under which anti-amyloid immunotherapy delivers benefits remains unclear^{34,35}. This is particularly pertinent as both Lecanemab and Donanemab had varying treatment effects based on the primary outcome in females, and, both Lecanemab and Donanemab showing no significant effects of treatment on the primary outcome in homozygotes for the *APOE-ε4* allele^{34 35}. Therefore, it is critical to quantify the impact genetic variation has on the cascading effects of Aβ on entorhinal tau accumulation and subsequent spread into the neocortex to best understand who to treat with these drugs.

Here, we use causal path modelling to assess how genetic variation impacts the AD pathological cascade (**Figure 1**). Using data from within subject multimodal PET and whole genome sequencing (WGS) in a sample of 1354 individuals we probe different stages of the AD cascade to understand how genetic variation in sex, *APOE-ε4* and *TREM2* exacerbate AD pathology. We tested the effect of genetic variation through pathways mediated by Aβ or via non-Aβ pathways, that is, primary tau accumulation in the EC and spread into the neocortex after accounting for Aβ. Our causal path is structured assuming that there are stages in the AD cascade, with the initial event being Aβ deposition, followed by Aβ-related deposition of tau in the EC, followed by tau spreading from the EC into the neocortex (**Figure 1**). We hypothesise that each genetic variable will enter the AD cascade through varied routes, interacting with either Aβ to increase levels of EC tau or with EC tau to increase levels of neocortical tau. Specifically, we hypothesise that sex will interact with Aβ to increase levels of primary tau and that the number of *APOE-ε4* alleles will have an Aβ independent effect on tau. Through more exploratory analyses, we hypothesise there will be variable and interactive effects of *TREM2* and *APOE-ε4* that increase tau spreading from primary regions into the neocortex (i.e. greater neocortical tau for a given level of primary tau).

And Discussion pg 12.

Here we used causal path analyses structured on the canonical AD pathological cascade where A β is the initiating event, followed by increased tau burden in the EC, followed by tau involvement of neocortex. Using this path framework, and genetic variations in sex, *TREM2* and *APOE- ϵ 4* as instrumental variables we can infer how different biological processes influence the AD cascade leading to increases in discrete stages of tau pathology (i.e., medial temporal tau and neocortical tau). We provide compelling evidence for heterogeneity in how regionally specific tau pathology is distributed based on different genetic traits, thus providing insight into the biological mechanisms that may govern increased tau pathology. Furthermore, we provide empirical evidence that may explain variable gene and sex related treatment effects of recent anti-amyloid immunotherapy trials.

*2) Neuropathology studies suggest that *TREM2* is more likely to result in an atypical AD presentation with more neocortical than medial temporal tau (10.1007/s00401-022-02495-4). In this study, a main effect of *TREM2* was seen in neocortical tau, but not entorhinal tau. Since these cortical-predominant cases may defy typical Braak-like progressions, it might be worthwhile to see whether a model where neocortical tau leads to entorhinal tau (rather than vice versa) is worth exploring, especially for *TREM2*. Does such a model fit the data as well or better in *TREM2* models? What about a model with bidirectional relationships between these two and simultaneous relationship with Abeta? These are plausible scenarios in cortical predominant cases.*

Here, we observe a significant interaction between EC tau and *TREM2* and a main effect of *TREM2* on higher levels of neocortical tau. Given the modelling architecture that includes both main and interactive terms they must be considered together. Our interpretation of these results is that *TREM2* interacts with primary tau accumulation in the EC to increase spread into the neocortex (i.e. in a Braak-like progression), while also having main effects on neocortical tau. As the reviewer suggested it is possible that that *TREM2* carriers are susceptible to tau accumulation in a non-Braak like topography with tau solely in the neocortex as has been reported previously

(10.1007/s00401-022-02495-4). However, when we investigate the marginal effects of EC tau on MetaTemp tau for *TREM2* carriers we observe a positive relationship between EC tau and MetaTemp tau, suggesting that the load of tau in the neocortex is not decoupled from the load of medial temporal tau as would be the case in a non Braak like pattern (i.e. no relationship between MetaTemp tau and EC tau in *TREM2* carriers). It is through these marginal effects that we do not believe that the results we are observing in *TREM2* carriers is evidence of non-braak like tau pathology (i.e. neocortical tau load in the absence of medial temporal tau).

We now mention this in the discussion pg17.

Third, our interpretation of the interactive effects of EC tau and *TREM2*, or *APOE-ε4* homozygosity on neocortical tau assumes typical Braak like progression (i.e. tau spreading from EC into neocortex), however it is plausible that our results could be consistent with a more atypical non-Braak like presentation of tau (i.e. high neocortical tau but no EC tau). Previous neuropathological work in patients with varied AD phenotypes suggests that *TREM2* variant carriers have a higher proportion of atypical, hippocampal sparing patterns of tau burden⁶⁴. However, the observation that in *TREM2* risk variant carriers there is still a positive interaction between EC-tau and MetaTemp tau levels suggests that the accumulation of tau in the neocortex is not entirely decoupled from the medial temporal lobes (i.e. hippocampal sparing).

In this work we only looked at up to two-way interactions due to sample size restrictions with *TREM2* carriers and *APOE4* homozygotes. To investigate the bidirectional and simultaneous relationship with Aβ would require the modelling of three-way interactions. With greater sample sizes for these rarer populations we hope to incorporate more complex interactive processes to further unpack the complex interactive processes that govern the spatial distribution and relative load of tau.

Finally, we include below (**Response to Reviewer 2, Table 1, Figure 1**) the suggested analysis reversing the order of the variables whereby MetaTemp tau now leads to EC tau. The interactive terms between *TREM2* and tau are consistent showing that there are lower levels of EC tau for a given level of MetaTemp tau.

Response to Reviewer 2, Table 1: Reversing EC tau and Meta Temp tau: the variable selection results and coefficient estimates in the discovery sample

	Estimate	p-value
Meta Temp tau as outcome		
(Intercept)	0.9845	<0.0001
$A\beta$	0.0016	<0.0001
Age	0.0019	0.0453
SEXfemale	0.0213	0.1017
$TREM2_1$	-0.0725	0.1763
$APOE-\epsilon 4_1$	-0.0016	0.9419
$APOE-\epsilon 4_2$	-0.1101	0.0305
$A\beta \times APOE-\epsilon 4_1$	0.0001	0.7813
$A\beta \times APOE-\epsilon 4_2$	0.0031	<0.0001
SEXfemale \times $TREM2_1$	0.1316	0.0154
$TREM2_1 \times APOE-\epsilon 4_1$	0.1552	0.0134
$TREM2_1 \times APOE-\epsilon 4_2$	-0.0809	0.4537
$A\beta \times TREM2_1$	-0.0014	0.0772
EC tau as outcome		
(Intercept)	-0.1182	0.0608
$A\beta$	<0.0001	0.9870
MetaTemp tau	0.9962	<0.0001
Age	0.0009	0.1228
SEXfemale	-0.0120	0.3047
$TREM2_1$	0.2877	0.0052
$APOE-\epsilon 4_1$	0.0256	0.6966
$APOE-\epsilon 4_2$	0.2856	0.0001
$40 < A\beta < 60 \times$ MetaTemp tau	0.0206	0.0145
MetaTemp tau \times $APOE-\epsilon 4_1$	-0.0005	0.9930
MetaTemp tau \times $APOE-\epsilon 4_2$	-0.1968	0.0016
MetaTemp tau \times $TREM2_1$	-0.2778	0.0012
$A\beta \times$ SEXfemale	0.0004	0.0573
SEXfemale \times $TREM2_1$	0.0604	0.0871

The following figures are similar to figure 3, but only focus on discovery sample and with MetaTemp tau and EC Tau reversed.

Response to Reviewer 2, Figure 1: Marginal effects reversing EC tau and Meta Temp tau:

3) Much of what was presented in the study is already well known or has been published. For instance, the idea that APOE leads to increased tau per level of Ab has been recently published (10.1001/jamaneurol.2023.4038). To this reviewer, the two most novel and innovative findings in the study are the findings related to the rare TREM2 mutations, and the integration of different variables into a large multi-variate directed model. However, both of these new and exciting findings have different flaws:

a) The TREM2 effects were all quite interesting, but were also the flimsiest findings in terms of support from the data. The samples were quite small (how large were the different sex groups of TREM2+ individuals?, or the E4 homozygote + TREM2 groups?), some of the effects were quite close to the alpha value for significance (e.g. the sex-TREM2 interaction on entorhinal tau) and, most importantly, there was no replication. As the least well understood element of the paper, this one is the most important to validate. I did not fully understand why these models were not attempted validation in HABS-HD. It seemed that there were TREM2 issues in all datasets, but they were determined too much to overcome in HABS-HD. It would be nice to see if the effects are similar even among subjects who did have TREM2 mutations in HABS-HD.

We had hoped to perform these validations in the HABS-HD sample, however of the whole genome sequencing (WGS) data from the HABS-HD study we found only 2 TREM2 SNPs were sampled. When these were combined into a risk carrier status it resulted in only 15 participants with TREM2 carrier risk status, or 2% of the HABS-HD sample, substantially lower than the 6%

within the discovery sample. Within these 15 HABS-HD participants there were only 4 *APOE-ε4* carriers with no homozygotes. Considering this, the data we had available in HABS-HD would not have permitted the inclusion of *APOE-ε4* homozygosity as a level of the *APOE-ε4* variable. The relatively low proportion of TREM2 risk carriers and the absence of TREM2 risk carriers who were *APOE-ε4* homozygotes in the HABS-HD sample reduced our confidence in the TREM2 indicator while also restricting our ability to model *APOE-ε4* homozygosity in the HABS-HD sample.

When we are testing interactions between TREM2 and other genetic variables (i.e. sex and *APOE-ε4*) our sample sizes for different levels within TREM2 carriers becomes small impacting statistical significance. Notwithstanding these concerns there was a statistical significance well below an alpha threshold of $p < 0.05$ for both the interactions between TREM2 and EC tau on MetaTemp tau as well as the main effect of TREM2 on MetaTemp tau after conditioning on EC tau. Thus, we still find our TREM2 findings a worthy and novel contribution to the literature on microglia and tau. However, we certainly agree that future work in additional samples is required to provide confidence in the TREM2 findings we presented in our discovery sample.

We have now included specific mention of both of these points, in particular the discussion now writes pg 18.

Finally, we were unable to run a replication analysis of TREM2 effects in the HABS-HD sample. This was due to the incomplete sampling of *TREM2* SNPs, limiting our ability to generate a reliable *TREM2* risk variant carrier status resulting in a substantially lower proportion of *TREM2* risk variant carriers than the discovery sample. This low proportion resulted in no *TREM2* risk variant carriers who were *APOE-ε4* homozygotes thus restricting our ability to appropriately model the data. Further work will be required to validate our *TREM2* findings, particularly those close to significance thresholds (i.e. interactions with sex and *APOE-ε4*). Notwithstanding, our findings regarding the main effects and interactions of *TREM2* with EC tau on neocortical tau were well below our significance threshold providing some confidence in these findings.

b) The final model in the paper was very interesting and produced novel results. The novelty comes from the approach of a multivariate directed framework. In that regard, what is the purpose of interpreting the first two models, which are smaller and represent a less complete

overall picture of the data? If results change after adding more variables and pathways, shouldn't this suggest that the earlier results were driven by these other explanatory data not being included?

We suspect that reviewers (and potential readers) will agree with us when we point out that this manuscript comprises a fairly complex set of models. To begin the results section with the final model would pose problems for a readers' ability to understand the framework adequately, since the upstream models both explain this framework and define the main effects and interaction effects that we combine in the final model presented in figure 4. Of note however, our results don't change due to the inclusion of different variables, rather it is the interpretation of the results of downstream models that are guided by the results of upstream models. It is in probing each stage of the AD cascade in this 3 staged approach that we can get the most complete overall picture of the data and cascading relationships between genetic variation and AD pathology.

3) Some of the APOE effects are puzzling. It seems that many of the effects described are not present in APOE e4 heterozygotes. For example, in Fig 3C, E4 heterozygotes are indistinguishable from controls. This is interesting because it is also more similar to many of the animal models that are described to support the APOE effect on tau, which tend to use homozygous expression, often on top of an AD genetic background. However, the APOE findings are presented in the discussion more broadly. Could the authors perhaps speculate as to why the effects do not appear to be dose dependent in their study, and how the reader should interpret this in the broader context of the APOE - tau relationship?

One must be careful about the hidden effect of APOE on disease progression. APOE carriers in the dataset are more likely to have amyloid, and to have MCI and AD, and these cases are all more likely to have more tau. The authors present an attempt at a causal model, but other factors could explain the findings (e.g. non-tau-mediated cognitive effects). The direct effect of APOE on tau would be made more solid by showing this trend is present just in Ab+ controls, and/or just in Ab+ MCI. Especially since these effects seem to contradict some other work out there suggesting amyloid mediates E4 effects on tau (e.g. 0.1007/s00259-021-05192-8).

In this work we observe a consistent dose dependent effect on the number of *APOE*- ϵ 4 alleles and A β burden. Then in subsequent downstream modelling we observe reliable and significant effects of A β as a predictor of EC tau and interacting with EC tau to predict MetaTemp tau. It is through these indirect (i.e. A β mediated) routes *APOE*- ϵ 4 has a dose dependent effect on tau. This mediation effect was reported in the above-mentioned work e.g. 0.1007/s00259-021-05192-8. However, a notable difference in this prior work to our current work is we do not aggregate *APOE*- ϵ 4 homozygotes and heterozygotes as a single population. By including *APOE*- ϵ 4 homozygotes as an independent population from heterozygotes we observe additional tau related effects of *APOE*- ϵ 4 that aren't mediated entirely through A β . In particular, we observe effects of *APOE*- ϵ 4

homozygosity on tau when conditioning on A β , or, the interaction of A β and EC tau. Thus, we show that *APOE*- ϵ 4 homozygosity increases tau spreading through interactions with primary tau (**Figure 3**), as well as increases primary tau through interactions with, and, mediated by A β (**Figure 2,4**).

We have now included as supplementary analysis models including cognitive status (control, MCI and AD) as a confounding variable. All *APOE*- ϵ 4 findings remain unchanged suggesting that our findings regarding *APOE*- ϵ 4 and tau are not driven by clinical stage.

In particular the discussion now writes pg 12

We observe strong effects of *APOE*- ϵ 4 homozygosity on both tau in EC and neocortex.

Specifically, across levels of A β *APOE*- ϵ 4 homozygotes had substantially more EC tau. Further, for a given level of EC tau *APOE*- ϵ 4 homozygotes had greater levels of neocortical tau after accounting for both direct and interactive effects of EC tau and A β . The net result of these effects means in comparison to *APOE*- ϵ 4 non-carriers and heterozygotes *APOE*- ϵ 4 homozygotes have a greater level of EC tau for a given level of A β , and, a greater level of neocortical tau for a given level of EC tau.

A notable result in this work is the relative absence of tau related effects for *APOE*- ϵ 4 heterozygotes. We observed a strong and reliable dose dependent effect of the number of *APOE*- ϵ 4 alleles on A β burden, with subsequent downstream modelling showing effects of A β on EC tau as well as interactive effects of A β and EC tau on MetaTemp tau. It is likely through these indirect A β mediated routes the number of *APOE*- ϵ 4 alleles has a dose dependent effect on tau. Our path modelling approach however conditions downstream models (i.e. with tau as an outcome) in a way to account for these A β mediated dose dependent effects, with the resultant effects of *APOE*- ϵ 4 homozygosity on tau over and above these A β mediated effects. Our analyses suggest that the sole

presence of the APOE4 isoform in humans has a strong effect on tau, both through interactions with existing A β pathology and interactions with primary tau pathology.

Multiple lines of evidence have shown that overexpression of APOE4 increases tau phosphorylation^{16,37} and spread of tau^{11,14}. In knock in mouse models with human *APOE- ϵ 4* homozygosity it is clear that only expressing the APOE4 isoform increases tau pathology through the activity of neuronal^{38,39}, astrocytic⁴⁰ and microglial^{41,42} cells. Prior work in these mouse models with blocked expression of APOE4 specifically in neuronal, or astrocytic cells resulted in a reduction of tau phosphorylation and spread compared to the mice solely expressing APOE4. Indicating that in the presence of only APOE4, tau proliferates to a greater extent. Furthermore, *APOE- ϵ 4* knock in mice have displayed hyperexcitability in medial temporal regions^{39,43}, a potential driver of tau accumulation⁴⁴.

Converging human neuroimaging studies have implicated the *APOE- ϵ 4* allele with increased levels of medial temporal tau independent of A β ⁴⁵⁻⁴⁸, although some reports do indicate that A β may mediate the relationship between *APOE- ϵ 4* and medial temporal tau. A likely influence of these discrepant results is the aggregation of *APOE- ϵ 4* carriership as a binary variable and not modelling *APOE- ϵ 4* homozygotes as a different population. Given the substantially greater risk of having clinical AD⁸ and A β positivity⁹ in homozygotes compared to heterozygotes, it is becoming clear that the statistical aggregation of *APOE- ϵ 4* homozygotes and heterozygotes in one group may not be appropriate⁹. Although we were well powered to make comparisons between *APOE- ϵ 4* homozygotes and heterozygotes, further work in larger samples will be required to fully understand the differences in tau burden between these groups.

Recent imaging studies have suggested that *APOE-ε4* potentiates the relationship between Aβ and tau pathologies ^{49,50}; our results provide some support of this model, highlighting that Aβ and *APOE-ε4* homozygosity interact to increase EC tau pathology. However, we suggest a refinement to this model whereby *APOE-ε4* homozygosity also potentiates the relationship between tau burden in the EC and spread into the neocortex through interactions with tau independent of Aβ.

Further, the results and supplementary now writes pg 7,8,10.

Furthermore, we observed negligible differences in estimated parameters for each sample when including diagnosis as a confounding variable.

Supplementary Table 4 Aβ as outcome, estimates in the discovery sample including diagnosis as confound.

	Estimate	p-value
(Intercept)	-34.3745	0.0619
Age	0.8937	0.0002
SEX(female)	-1.5275	0.6153
TREM2(1)	-0.9433	0.8765
APOE-ε4(1)	26.3210	0.0000
APOE-ε4(2)	41.7012	0.0000
DIAG(Dementia)	24.3724	0.0007
DIAG(MCI)	-10.8320	0.0120

Supplementary Table 4: Estimations and p-values of the coefficients in *Aβ* response model

Supplementary Table 5 Aβ as outcome, estimates in the replication sample including diagnosis as confound.

	Estimate	p-value
(Intercept)	-70.9176	0.0000
Age	1.1622	0.0000
SEX(female)	6.2101	0.0014
APOE-ε4(1)	11.6513	0.0000
APOE-ε4(2)	24.0164	0.0000
DIAG(Dementia)	13.1098	0.0008
DIAG(MCI)	3.2123	0.1921

Supplementary Table 5: Estimations and p-values of the coefficients in $A\beta$ response model

Table 9 EC tau as outcome, estimates in the discovery sample including diagnosis as confound.

	Estimate	p-value
(Intercept)	1.0372	0.0000
$A\beta$	0.0012	0.0000
Age	0.0004	0.7109
SEX(female)	-0.0008	0.9704
TREM2(1)	-0.0647	0.1631
APOE- ϵ 4(1)	0.0309	0.1896
APOE- ϵ 4(2)	-0.0249	0.6516
DIAG-Dementia	0.2292	0.0000
DIAG-MCI	0.0680	0.0006
$A\beta \times$ APOE- ϵ 4(1)	0.0001	0.8919
$A\beta \times$ APOE- ϵ 4(2)	0.0017	0.0116
$A\beta \times$ SEX(female)	0.0008	0.0144
SEX(female) \times TREM2(1)	0.1061	0.0664

Supplementary Table 9: Estimations and p-values of the coefficients in EC tau response model

Table 10 EC tau as outcome, estimates in the replication sample including diagnosis as confound.

	Estimate	p-value
(Intercept)	1.2575	0.0000
$A\beta$	0.0020	0.0002
Age	0.0003	0.7937
SEX(female)	-0.0209	0.2494
APOE- ϵ 4(1)	0.0207	0.3319
APOE- ϵ 4(2)	-0.0557	0.3992
DIAG(Dementia)	0.0857	0.0102
DIAG(MCI)	0.0253	0.2247
$A\beta \times$ APOE- ϵ 4(1)	0.0006	0.3037
$A\beta \times$ APOE- ϵ 4(2)	0.0035	0.0236
$A\beta \times$ SEX(female)	0.0012	0.0386

Supplementary Table 10: Estimations and p-values of the coefficients in EC tau response model

Supplementary Table 14 MetaTemp tau as outcome, estimates in the discovery sample including diagnosis as confound.

	Estimate	p-value
(Intercept)	0.4638	0.0000
A β	0.0002	0.1324
EC tau	0.6611	0.0000
Age	-0.0006	0.2474
SEX(female)	0.0585	0.1608
TREM2(1)	-0.3302	0.0002
APOE- ϵ 4(1)	-0.0210	0.6495
APOE- ϵ 4(2)	-0.4540	0.0000
A β >60 \times EC tau	0.0168	0.0865
DIAG(Dementia)	0.0671	0.0001
DIAG(MCI)	0.0116	0.2402
EC tau \times APOE- ϵ 4(1)	0.0025	0.9489
EC tau \times APOE- ϵ 4(2)	0.3545	0.0000
EC tau \times TREM2(1)	0.2854	0.0002
TREM2(1) \times APOE- ϵ 4(1)	0.0768	0.0237
TREM2(1) \times APOE- ϵ 4(2)	-0.0055	0.9246
A β \times TREM2(1)	-0.0006	0.1894
EC tau \times SEX(female)	-0.0454	0.1958

Supplementary Table 14: Estimations and p-values of the coefficients in MetaTemp tau response model

Supplementary Table 15 MetaTemp tau as outcome, estimates in the replication sample including diagnosis as confound.

	Estimate	p-value
(Intercept)	0.5392	0.0000
A β	-0.0001	0.8190
EC tau	0.4405	0.0000
Age	-0.0004	0.3181
SEX(female)	-0.0177	0.6547
APOE- ϵ 4(1)	0.1013	0.0158
APOE- ϵ 4(2)	-0.1759	0.0259
A β \geq 60 \times EC tau	0.0929	0.0000
DIAG(Dementia)	0.0736	0.0000
DIAG(MCI)	0.0060	0.4810
EC tau \times APOE- ϵ 4(1)	-0.0809	0.0085
EC tau \times APOE- ϵ 4(2)	0.1267	0.0206

EC tau×SEX(female)	0.0360	0.2224
--------	--------

Supplementary Table 15: Estimations and p-values of the coefficients in MetaTemp tau response model

4) Causality is a polarizing term in this context. The authors are certainly using causal models, and there is certainly a hypothesis of implied causality between variables in this dataset. But to say that the authors are reporting “causal effects” is probably overblown. We don’t know if the effects are causal, even if this causal model supports the claim that they are. I would recommend removing phrases and section headings that purport causal effects, or at least amend them to say “purported causal effects” or something like that. In addition, it is not entirely clear how the models are chosen, and whether alternative models better support the data.

We have now amended our description of the effects omitting the term causal in their description. We still refer to the models as causal for statistical brevity.

5) The replication sample is a huge strength of the study and many of the replications are impressive. I am curious of the authors’ perspective why, for the model described in Tables S1 and S2, the APOE effect size is cut in half in the HABS-HD dataset compared to the discovery dataset.

The difference in effect size is due to the differences in the populations studied. The HABS-HD sample is younger and less impaired i.e. likely to be earlier along the AD cascade. These factors lead to a different distribution of amyloid levels in the sample which resulted in attenuated effect sizes for *APOE-ε4*. We believe replicating in this heterogeneous sample relative to the discovery sample is a major strength of this study highlighting the reliability of our findings.

6) Figure 1 is a nice description of all of the relationships in this study. Still, it would be useful for each figure thereafter, a similar diagram is given highlighting the relationships involved. I thought that was there but those figures do not depict any interactions/mediations that are clearly involved in those models, so its a bit hard to understand how they relate.

We have now included each path modelled in each section including main and interactive effects that are modelled. Further, we provide a summary of the influences of each genetic variable on the accumulation of AD pathologies.

Figure 1 Canonical Alzheimer's disease cascade. The left panel indicates the stages and spatial distribution of Alzheimer's Disease (AD) pathology throughout the cascade. We modelled the initial stages of tau pathology using the entorhinal cortex (Tau_{EC}) region of interest. Neocortical

tau was modelled using the meta temporal (Tau_{Meta}) region of interest. The right panel shows the directed acyclic graph used to model the potential pathways between genetic variables and pathology. Each line originates at a predictor variable with the yellow nodes indicating an AD pathology. Solid lines indicate the pathway from an upstream variable to a downstream pathology, dashed lines indicate an interactive effect between upstream variables on a downstream pathology, with the arrow indicating a modulation of the pathway showing by the solid line. Solid and dashed lines initiating from the black genetic variables box indicate a direct or interactive effect respectively of genetic variable on the outcome variable. The black box indicates the three genetic variables that are modelled as direct or interactive predictors of AD pathologies, these are the number of Apolipoprotein E (*APOE*)- $\epsilon 4$ alleles (0,1,2), Sex (female, male), and Triggering Receptor Expressed on Myeloid Cells 2 (*TREM2*) risk variant carrier status (0,1).

Discovery Sample

Replication Sample

Figure 2 Genetic influences on entorhinal cortex tau. Lines show the estimated marginal levels of tau pathology for individuals with varying genetic profiles at different levels of amyloid beta ($A\beta$). Whisker plot shows estimated levels of entorhinal cortex tau (Tau_{EC}) pathology based on sex and Triggering Receptor Expressed on Myeloid Cells 2 (*TREM2*) risk variant carrier status. Different effects in the discovery sample for **a.** Apolipoprotein E (*APOE*)- $\epsilon 4$, **b.** sex, **c.** interaction of sex and *TREM2* risk variant carrier status. Different effects in the replication sample for **d.** *APOE*- $\epsilon 4$ and **e.** sex. **f.** Causal path model to estimate Tau_{EC} . Solid lines indicate a direct effect of an upstream variable on a downstream pathology, dashed lines indicate an interactive effect between upstream variables (i.e. $A\beta$ centiloid or genetic variable) on Tau_{EC} . Solid lines initiating from the black genetic variables box may indicate a direct or interactive effect of genetic variables on the outcome variable. Values in the parentheses indicate level of genetic variable (*APOE*- $\epsilon 4$ alleles (0,1,2) and *TREM2* risk variant carrier status (0,1)).

Figure 3 Genetic influences on meta temporal tau. Lines show the estimated marginal levels of meta temporal tau (Tau_{Meta}) pathology for individuals with varying genetic profiles at different levels of entorhinal cortex tau (Tau_{EC}). Different effects in the discovery sample for **a.** Apolipoprotein E ($APOE$)- $\epsilon 4$, **b.** Triggering Receptor Expressed on Myeloid Cells 2 ($TREM2$), and **c.** their interaction. Different effects in the replication sample for **d.** $APOE$ - $\epsilon 4$. **e.** Causal path model to estimate Tau_{Meta} . Solid lines indicate a direct effect of an upstream variable on a downstream pathology, dashed lines indicate an interactive effect between upstream variables (i.e. amyloid beta ($A\beta$), Tau_{EC} , or genetic variable) on meta temporal tau. Solid lines initiating from the black genetic variables box may indicate a direct or interactive effect of genetic variables on the outcome variable. Values in the parentheses indicate level of genetic variable ($APOE$ - $\epsilon 4$ alleles (0,1,2) and $TREM2$ risk variant carrier status (0,1)).

Figure 5 Genetic influences on the AD cascade. Causal path models indicating significant pathways in which genetic variation leads to increased Alzheimer's Disease (AD) pathology. Each line originates at a genetic variable with the yellow nodes indicating an outcome variable. Solid lines indicate the pathway from an upstream variable to a downstream pathology, dashed lines indicate an interactive effect between upstream variables on a downstream pathology with the arrow indicating a modulation of the pathway showing by the solid line. a. effect of Apolipoprotein E (*APOE*)- $\epsilon 4$ homozygosity on AD pathology, b. effect of Triggering Receptor Expressed on Myeloid Cells 2 (*TREM2*) risk variant carrier status on AD pathology, c. effect of female sex on AD pathology, d. interactive effect of *APOE*- $\epsilon 4$ and *TREM2* risk variant carrier status on AD pathology, and e. effect of female sex and *TREM2* risk variant carrier status on AD pathology.

7) No data is shown and it would be nice to at least chart confidence intervals in the Figures showing marginal relationships. Figure 2C is even a barplot. I'm pretty sure Nat Comms doesn't even allow barplots, and for good reason.

We now present in supplementary each marginal relationship with confidence intervals. We present each relationship as an independent panel for clarity and consistent presentation to ensure interpretability when many marginal effects are shown together (e.g., Figure 3 d). Further we have now removed the barplot in Figure 2C, replacing this with a whisker plot showing the estimated levels and confidence intervals of EC tau for different levels of the *TREM2* and sex interaction.

Supplementary Figure 1 Genetic influences on Entorhinal Cortex (EC) tau. Lines show the estimated marginal levels of entorhinal cortex (Tau_{EC}) pathology for individuals with varying genetic profiles at different levels of $A\beta$, shaded portions represent the confidence intervals for the effect size across different levels of $A\beta$. Different effects in the discovery sample for **a.-c.** *APOE- ϵ 4*, **d.,e.** sex. Different effects in the replication sample for **f.-h.** *APOE- ϵ 4* and **i.,j.** sex. Values in the parentheses indicate level of genetic variable (*APOE- ϵ 4* alleles (0,1,2) and *TREM2* risk variant carrier status (0,1)).

Discovery Sample

Replication Sample

Supplementary Figure 2 Genetic influences on meta temporal tau. Lines show the estimated marginal levels of meta temporal (Tau_{Meta}) pathology for individuals with varying genetic profiles at different levels of entorhinal cortex tau (Tau_{EC}), shaded portions represent the confidence intervals for the effect size across different levels of Tau_{EC} . Different effects in the discovery sample for **a.,b.** *TREM2*, **c.-e.** *APOE- ϵ 4*, **f.-k.** interaction of *TREM2* and *APOE- ϵ 4*. Different effects in the replication sample for **i.-n.** *APOE- ϵ 4*. Values in the parentheses indicate level of genetic variable (*APOE- ϵ 4* alleles (0,1,2) and *TREM2* risk variant carrier status (0,1)).

8) While certainly interesting, I have trouble interpreting the effect of female sex on entorhinal but not cortical tau in the full multivariate model toward the end of the paper. The author's seem to interpret this as vulnerability, but could this not also be interpreted as resistance? In other words (if we believe the models here), that women are able to retain more pathology in the entorhinal cortex before it spreads out into neocortex?

The constructs of both resistance and resilience are best tested in an interaction model, where the amount of a dependent variable C is affected by an effect modifier B on the independent variable A⁵¹. Resistance to the effects of primary tau in woman would manifest in our models as an interaction between EC tau and sex(females) when predicting MetaTemp tau, whereby women would have lower neocortical tau for a given level of EC tau (i.e. more resistant to the spreading pathways of tau than men). We did not observe this and thus we do not believe our data supports a finding that females are able to retain more tau in the EC before it spreads into the neocortex. However, the modification of the relationship between $\text{A}\beta$ and EC tau by sex is consistent with the opposite pattern, suggesting vulnerability.

9) Can the authors explain the binning windows of centiloids? Much has been made about logical cut-offs for centiloids. With reference to this work, wouldn't it make more sense for the bins to be 0-20, 20-40, 40-60, >60? Do the amyloid interactions change at all using this binning strategy?

In the original model, we chose bins for $\text{A}\beta$ as <10, 10-40, 40-60, and >60 to allow for better interpretation of the $\text{A}\beta$ *EC Tau interaction term and to facilitate mediation effect calculations. After applying stepwise variable selection via AIC, only the $\text{A}\beta$ *Tau EC >60 interaction was retained in the final model.

We tested the alternative binning strategy you suggested (<20, 20-40, 40-60, >60) and applied the same variable screening procedure. The results remained consistent with the original approach, with only the $\text{A}\beta$ *Tau EC >60 interaction term being retained in the final model.

Therefore, we conclude that adopting the proposed binning strategy does not impact the variable selection or the regression results in our analysis.

*10) Conceptually I have some trouble fully understanding the proposed causality of TREM2 in the models. Genetic markers are obviously upstream of measured phenotypes. With APOE, there is strong evidence supporting the idea that its effects are causally driving amyloid accumulation. However, most data and hypotheses purport microglia effects to become relevant *after* Ab deposition occurs. If this is the case, and if the authors agree, would it not make more sense to generate some sort of latent TREM2 phenotype (e.g. microglial response) that is influenced by (e.g. downstream of) TREM2 but also downstream of Abeta? If such a factor, or TREM2 itself, is placed after Ab, does it the model fit the data similarly or better?*

Within our modelling framework we are able to infer at which stage a genetic variable may become relevant (i.e. before/after A β accumulation, or before/after EC tau accumulation). Our findings showing an interaction between *TREM2* and EC tau on MetaTemp tau which was significant after conditioning on A β . This suggests -as the reviewer has highlighted- that *TREM2* effects may be downstream of A β . We present this finding in the context of the biological role of *TREM2* in microglial function, whereby the response of microglial to primary tau exacerbates MetaTemp tau. In the initial model predicting A β *TREM2* was not a predictor of A β . However, it was a significant predictor when modelling effects downstream of A β (i.e. predicting EC and MetaTemp tau) in general agreement with the reviewers comments. In particular we observed that *TREM2* impacts tau through interactions with sex and primary tau -when predicting neocortical tau-. It may be feasible to include s*TREM2* from fluid biomarkers and incorporate this in the path modelling framework as a response phenotype downstream to both *TREM2* and A β but this falls beyond the remit of this work.

Reviewer #3

In this manuscript, Giorgio et al. investigated how genetic variation (here reflected by sex, APOE e4 dosage, and TREM2 risk variant carriership) moderated different aspects of the amyloid cascade hypothesis (here reflected by amyloid deposition leading to entorhinal tau leading to neocortical tau). The manuscript is well written and overall clear, and the use of a racially diverse replication cohort strengthens their findings. However, there are also some limitations to this work among which the lack of longitudinal data and the pooling of participants of different cognitive stages. In addition, the final part of the results section was relatively difficult to grasp.

I have the following questions, comments and suggestions:

1. The cohort includes a mixture of CU, MCI and dementia participants and the authors do not control for this in their analyses. I think it would be of importance to show that the observed associations are not driven by differences in disease stage.

We now include as supplement models that include clinical disease stage as a covariate resulting in no substantial changes to the results or model interpretation. We note that estimates are highly similar although there is a slight attenuation in statistical power when investigating the less frequent combinations of genetic risk.

In particular the results and supplementary now writes pg 7,8,10.

Furthermore, we observed negligible differences in estimated parameters for each sample when including diagnosis as a confounding variable.

Supplementary Table 4 $A\beta$ as outcome, estimates in the discovery sample including diagnosis as confound.

	Estimate	p-value
(Intercept)	-34.3745	0.0619
Age	0.8937	0.0002
SEX(female)	-1.5275	0.6153
TREM2(1)	-0.9433	0.8765
APOE- ϵ 4(1)	26.3210	0.0000
APOE- ϵ 4(2)	41.7012	0.0000
DIAG(Dementia)	24.3724	0.0007
DIAG(MCI)	-10.8320	0.0120

Supplementary Table 4: Estimations and p-values of the coefficients in $A\beta$ response model

Supplementary Table 5 $A\beta$ as outcome, estimates in the replication sample including diagnosis as confound.

	Estimate	p-value
(Intercept)	-70.9176	0.0000
Age	1.1622	0.0000
SEX(female)	6.2101	0.0014
APOE- ϵ 4(1)	11.6513	0.0000
APOE- ϵ 4(2)	24.0164	0.0000
DIAG(Dementia)	13.1098	0.0008
DIAG(MCI)	3.2123	0.1921

Supplementary Table 5: Estimations and p-values of the coefficients in $A\beta$ response model

Table 9 EC tau as outcome, estimates in the discovery sample including diagnosis as confound.

	Estimate	p-value
(Intercept)	1.0372	0.0000
$A\beta$	0.0012	0.0000
Age	0.0004	0.7109
SEX(female)	-0.0008	0.9704
TREM2(1)	-0.0647	0.1631
APOE- ϵ 4(1)	0.0309	0.1896
APOE- ϵ 4(2)	-0.0249	0.6516
DIAG-Dementia	0.2292	0.0000
DIAG-MCI	0.0680	0.0006
$A\beta \times$ APOE- ϵ 4(1)	0.0001	0.8919
$A\beta \times$ APOE- ϵ 4(2)	0.0017	0.0116
$A\beta \times$ SEX(female)	0.0008	0.0144
SEX(female) \times TREM2(1)	0.1061	0.0664

Supplementary Table 9: Estimations and p-values of the coefficients in EC tau response model

Table 10 EC tau as outcome, estimates in the replication sample including diagnosis as confound.

	Estimate	p-value
(Intercept)	1.2575	0.0000
$A\beta$	0.0020	0.0002
Age	0.0003	0.7937
SEX(female)	-0.0209	0.2494
APOE- ϵ 4(1)	0.0207	0.3319

APOE- ϵ 4(2)	-0.0557	0.3992
DIAG(Dementia)	0.0857	0.0102
DIAG(MCI)	0.0253	0.2247
A β ×APOE- ϵ 4(1)	0.0006	0.3037
A β ×APOE- ϵ 4(2)	0.0035	0.0236
A β ×SEX(female)	0.0012	0.0386

Supplementary Table 10: Estimations and p-values of the coefficients in EC tau response model

Supplementary Table 14 MetaTemp tau as outcome, estimates in the discovery sample including diagnosis as confound.

	Estimate	p-value
(Intercept)	0.4638	0.0000
A β	0.0002	0.1324
EC tau	0.6611	0.0000
Age	-0.0006	0.2474
SEX(female)	0.0585	0.1608
TREM2(1)	-0.3302	0.0002
APOE- ϵ 4(1)	-0.0210	0.6495
APOE- ϵ 4(2)	-0.4540	0.0000
A β >60 × EC tau	0.0168	0.0865
DIAG(Dementia)	0.0671	0.0001
DIAG(MCI)	0.0116	0.2402
EC tau × APOE- ϵ 4(1)	0.0025	0.9489
EC tau × APOE- ϵ 4(2)	0.3545	0.0000
EC tau × TREM2(1)	0.2854	0.0002
TREM2(1) × APOE- ϵ 4(1)	0.0768	0.0237
TREM2(1) × APOE- ϵ 4(2)	-0.0055	0.9246
A β × TREM2(1)	-0.0006	0.1894
EC tau × SEX(female)	-0.0454	0.1958

Supplementary Table 14: Estimations and p-values of the coefficients in MetaTemp tau response model

Supplementary Table 15 MetaTemp tau as outcome, estimates in the replication sample including diagnosis as confound.

	Estimate	p-value
(Intercept)	0.5392	0.0000
A β	-0.0001	0.8190

EC tau	0.4405	0.0000
Age	-0.0004	0.3181
SEX(female)	-0.0177	0.6547
APOE- ϵ 4(1)	0.1013	0.0158
APOE- ϵ 4(2)	-0.1759	0.0259
A β \geq 60 \times EC tau	0.0929	0.0000
DIAG(Dementia)	0.0736	0.0000
DIAG(MCI)	0.0060	0.4810
EC tau \times APOE- ϵ 4(1)	-0.0809	0.0085
EC tau \times APOE- ϵ 4(2)	0.1267	0.0206
EC tau \times SEX(female)	0.0360	0.2224

Supplementary Table 15: Estimations and p-values of the coefficients in MetaTemp tau response model

*2. One of the limitations of this study is its cross-sectional nature (while the AD/amyloid cascade is a long longitudinal process). Previous studies have been published performing comparable cross-sectional models, e.g. looking at effects of sex*amyloid on tau (Buckley et al. 2019 JAMA Neurology), or APOE*amyloid on tau (Therriault et al. 2020 Molecular Psychiatry). One of the strengths of the current study is that sex, APOE and TREM2 are modelled simultaneously thereby investigating their independent effects. However, if the authors additionally have longitudinal tau-PET data at hands (which at least should be available in ADNI), it could be of interest to include that here in order to further understand the influence of genetic variation on the longitudinal progression of tau.*

Our subject selection in ADNI was highly restricted due to the requirement of whole genome sequencing (WGS) to assign TREM2 risk carrier status, this limits the sample size with longitudinal tau PET in ADNI. We found that of those in our sample with WGS, 184 participants had 2 or more tau pet sessions (as of November 2024). Of these 184 participants 12 of them were TREM2 risk variant carriers and none of these 12 participants were APOE- ϵ 4 homozygotes. In the absence of this level (i.e. TREM2 carriers who are APOE- ϵ 4 homozygous) we cannot specify an interactive model. We hope in the future either more longitudinal tau PET data or WGS will become available to test these interactive pathways in the future.

3. The final part of the results section (“Variable genetic effects at different stages in the canonical amyloid cascade pathway”) is difficult to grasp. Very few statistics/results are reported in the text, and there is no accompanying visualization that can guide the reader. It would be helpful if the authors can add a figure visualizing all pathways and if the authors can

include the strengths/significances of all pathways (all direct effects, mediation effects and total effects – for all genetic groups) (also see point #6).

We have now re-written the final section and included a figure of the total, direct and mediation effects to improve interpretability. The new figure shows the 95% bootstrap confidence interval for the effect difference across different genetic groups within the replication sample. Since $A\beta$ does not interact with genetic factors in the MetaTemp Tau response model, the direct effect of $A\beta$ on MetaTemp Tau is a constant across different genetic groups. Consequently, the effect difference among these groups will always be zero. Therefore, we exclude the direct effect of $A\beta$ on MetaTemp Tau from the subsequent plot.

Variable genetic effects at different stages in the canonical amyloid cascade pathway.

Finally, we investigated the variability in the direct, mediation, and total effects of upstream pathology on downstream tau pathology for different genetic profiles. This allows us to probe each aspect of the AD cascade and estimate which groups are likely to have higher downstream pathology for a given level of upstream pathology (i.e. if a population has greater EC tau for a given level of $A\beta$ and how this may result in higher levels of MetaTemp tau). To do this, in each group we calculate the direct effect of $A\beta$ on EC tau ($DE_{A\beta \rightarrow EC \text{ tau}}$ 4a); the direct effect of EC tau on MetaTemp tau ($DE_{EC \text{ tau} \rightarrow MetaTemp \text{ tau}}$ 4b); the direct effect of $A\beta$ on MetaTemp tau ($DE_{A\beta \rightarrow MetaTemp \text{ tau}}$); the mediation effect of $A\beta$ through EC tau on MetaTemp tau ($ME_{A\beta \rightarrow EC \text{ tau} \rightarrow MetaTemp \text{ tau}}$ 4c), which is the product between the direct effect of EC tau on MetaTemp tau and the direct effect of $A\beta$ on EC tau; and finally the total effect of $A\beta$ on MetaTemp tau ($TE_{A\beta \rightarrow MetaTemp \text{ tau}}$ 4d), which is calculated as the direct sum of the mediation effect through the path $ME_{A\beta \rightarrow EC \text{ tau} \rightarrow MetaTemp \text{ tau}}$ and the direct effect of $A\beta$ on MetaTemp tau. Since $A\beta$ does not interact with genetic factors in the MetaTemp Tau response model, the direct effect of $A\beta$ on MetaTemp Tau is a constant across different genetic groups. Consequently, the effect difference among these groups will always be zero.

We observed that the total effect of A β on MetaTemp tau differs by APOE- ϵ 4 allele status. In particular, APOE- ϵ 4 homozygotes have significantly higher MetaTemp tau levels at a given level of A β compared to individuals with zero or one ϵ 4 allele ($TE_{A\beta \rightarrow \text{MetaTemp tau}}$: APOE- ϵ 4 (2 alleles) vs. (0 alleles)=0.003, $p=0.022$; APOE- ϵ 4 (2 alleles) vs. (1 allele)=0.0029, $p=0.035$; see **Figure 4d**, **Supplementary Table 16**). This difference is driven by the mediation pathway from A β to EC tau and subsequently to MetaTemp tau (**Figure 4c**). Because the mediation effect through this pathway is the product of the direct effect of A β on EC tau and the direct effect of EC tau on MetaTemp tau (**Figures 4a–b**), these findings imply that in APOE- ϵ 4 homozygotes, a given level of A β leads to higher EC tau, and a given level of EC tau leads to higher MetaTemp tau. Consequently, this population shows elevated MetaTemp tau at a given level of A β due to enhanced A β -driven EC tau aggregation and enhanced EC tau propagation into the neocortex.

There were no significant differences in the $TE_{A\beta \rightarrow \text{MetaTemp tau}}$ across other genetic groups, but we note that several numerical differences were present (albeit not reliably statistically significant). In the TREM2 group, the $TE_{A\beta \rightarrow \text{MetaTemp tau}}$ was not significant; however, the mediation effect through A β and EC tau (i.e., A $\beta \rightarrow$ EC tau \rightarrow MetaTemp tau) was significant ($TREM2(1)$ vs $TREM2(0)$ = 0.0008, $p = 0.014$). This was driven by significant differences in the direct effect of A β on EC tau ($TREM2(1)$ vs $TREM2(0)$ = 0.0002, $p = 0.027$) and in the direct effect of EC tau on MetaTemp tau ($TREM2(1)$ vs $TREM2(0)$ = 0.3145, $p = 0.029$) (**Supplementary Table 16**). These findings suggest that, in TREM2 risk variant carriers, across levels of A β there are higher levels of EC tau, and in turn, a given level of EC tau leads to higher MetaTemp tau. Although the mediation effect is significant, the overall total effect is diminished by the low-powered direct effect of A β

on MetaTemp tau, likely due to the limited sample size of *TREM2* risk variant carriers in the discovery cohort.

For females vs males comparison there was a non-significant numerical difference in the TEA β - > MetaTemp tau (females vs males=0.0005, p=0.12), which was driven by significant differences in the DE $A\beta$ -> EC tau (females vs males=0.0008, p=0.036) (**Supplementary Table 16**). This implies that for a given level of A β , women have greater levels of EC tau, and a trend towards greater MetaTemp tau for the same level of A β , a finding which was supported in the HABS-HD replication sample. Relative effects and interpretations for other contrasts were similar in the replication sample (**Supplementary Table 17, Figure 4 e-h**).

Figure 4 Direct, mediation and total effects of upstream pathology on tau for different levels of sex, APOE-ε4, and TREM2 risk carrier status. In the discovery sample (top panels a–d) and the replication sample (bottom panels e–h), panels a and e show the direct effect from amyloid beta (Aβ) red; entorhinal cortex tau (Tau_{EC}); panels b and f show the direct effect from Tau_{EC} to meta temporal tau (Tau_{Meta}); panels c and g show the mediation effect along the path Aβ → Tau_{EC} → Tau_{Meta} tau; and panels d and h show the total effect from Aβ to Tau_{Meta}.

4. Regarding the final part of the results: is it correct that TE reflects DE + ME? If so, I would specify that as this is currently not entirely clear.

Yes. The total effect (TE) is a direct sum of direct effect (DE) and mediation effect (ME) in each path. We have edited the description in the resubmission.

In particular the results now writes pg 10

total effect of A β on MetaTemp tau ($TE_{A\beta \rightarrow \text{MetaTemp tau}}$), which is calculated as the direct sum of the mediation effect through the path $ME_{A\beta \rightarrow \text{EC-tau} \rightarrow \text{MetaTemp tau}}$ and the direct effect of A β on MetaTemp tau.

5. Regarding the final part of the results: I found this sentence complex “When we contrast these effects amongst [...] (2-alleles) vs (1-allele)=0.0029,p=0.0335).” Please add the comparison group in the sentence (i.e., between APOE- ϵ 4 homozygotes and ...?), and there seems to be a bracket missing in that sentence. In addition, what do the 0.003 and 0.0029 values precisely reflect? From the text it seems to reflect the difference in the total effect, but from Supplementary Table 10 it seems to reflect the difference in the mediation effect. Also, does this reflect a difference in a standardized beta or an unstandardized beta?

We have now edited this section improving clarity. The differences are unstandardized betas and as such their scale represents either the range of Centiloids or tau PET SUVR values.

6. Regarding the final part of the results: in the text it is stated there are ‘no significant differences in the TE across the other genetic groups’, but in Supplementary Table 10 there does seem to be an effect for TREM2.

We have now edited this section improving clarity and have now included mention of the effects of TREM2.

7. Regarding the causal path models displayed in Figures 2-3: it would be helpful to include the path coefficients (numbers along the arrows representing the strengths and significances of the relationships) in order to visualize which paths are significant and which are not.

We have now included a summary figure (**Figure 5**) with the pathways that are significantly different based on genetic variation. As the path coefficients are different in the two samples we do not include the strengths, rather we just present the statistically significant paths as arrows.

Figure 5 Genetic influences on the AD cascade. Causal path models indicating significant pathways in which genetic variation leads to increased Alzheimer’s Disease (AD) pathology. Each line originates at a genetic variable with the yellow nodes indicating an outcome variable. Solid lines indicate the pathway from an upstream variable to a downstream pathology, dashed lines indicate an interactive effect between upstream variables on a downstream pathology with the arrow indicating a modulation of the pathway showing by the solid line. a. effect of Apolipoprotein E (*APOE*)- $\epsilon 4$ homozygosity on AD pathology, b. effect of Triggering Receptor Expressed on Myeloid Cells 2 (*TREM2*) risk variant carriership on AD pathology, c. effect of female sex on AD pathology, d. interactive effect of *APOE*- $\epsilon 4$ and *TREM2* risk variant carrier status on AD pathology, and e. effect of female sex and *TREM2* risk variant carrier status on AD pathology.

8. Regarding the causal path models displayed in Figures 1-2-3: while stated in the legend, I am not sure whether these models are currently visualizing any interaction effects. I think the causal path models are currently only visualizing main effects of *APOE*/sex/*TREM2* on amyloid/tau. To also visualize interaction effects within these causal path models, I think one would also need to add arrows going from e.g. *APOE* towards the arrow going from *AB*>*TAU* (in case one would want to visualize the interaction effect of *APOE***AB* on tau).

We have now updated the path diagrams to include interaction effects (as dashed lines).

Figure 1 Canonical Alzheimer’s disease cascade. The left panel indicates the stages and spatial distribution of Alzheimer’s Disease (AD) pathology throughout the cascade. We modelled the initial stages of tau pathology using the entorhinal cortex (Tau_{EC}) region of interest. Neocortical tau was modelled using the meta temporal (Tau_{META}) region of interest. The right panel shows the directed acyclic graph used to model the potential pathways between genetic variables and pathology. Each line originates at a predictor variable with the yellow nodes indicating an AD pathology. Solid lines indicate the pathway from an upstream variable to a downstream pathology, dashed lines indicate an interactive effect between upstream variables on a downstream pathology, with the arrow indicating a modulation of the pathway showing by the solid line. Solid and dashed lines initiating from the black genetic variables box indicate a direct or interactive effect respectively of genetic variable on the outcome variable. The black box indicates the three genetic variables that are modelled as direct or interactive predictors of AD pathologies, these are the number of Apolipoprotein E (*APOE*)- $\epsilon 4$ alleles (0,1,2), Sex (female, male), and Triggering Receptor Expressed on Myeloid Cells 2 (*TREM2*) risk variant carrier status (0,1).

Figure 2 Genetic influences on entorhinal cortex tau. Lines show the estimated marginal levels of tau pathology for individuals with varying genetic profiles at different levels of amyloid beta ($A\beta$). Whisker plot shows estimated levels of entorhinal cortex tau (Tau_{EC}) pathology based on sex and Triggering Receptor Expressed on Myeloid Cells 2 ($TREM2$) risk variant carrier status. Different effects in the discovery sample for **a.** Apolipoprotein E ($APOE$) - $\epsilon 4$, **b.** sex, **c.** interaction of sex and $TREM2$ risk variant carrier status. Different effects in the replication sample for **d.** $APOE$ - $\epsilon 4$ and **e.** sex. **f.** Causal path model to estimate Tau_{EC} . Solid lines indicate a direct effect of an upstream variable on a downstream pathology, dashed lines indicate an interactive effect between upstream variables (i.e. $A\beta$ centiloid or genetic variable) on Tau_{EC} . Solid lines initiating from the black genetic variables box may indicate a direct or interactive effect of genetic variables on the outcome variable. Values in the parentheses indicate level of genetic variable ($APOE$ - $\epsilon 4$ alleles (0,1,2) and $TREM2$ risk variant carrier status (0,1)).

Figure 3 Genetic influences on meta temporal tau. Lines show the estimated marginal levels of meta temporal tau (Tau_{Meta}) pathology for individuals with varying genetic profiles at different levels of entorhinal cortex tau (Tau_{EC}). Different effects in the discovery sample for **a.** Apolipoprotein E (*APOE*)-ε4, **b.** Triggering Receptor Expressed on Myeloid Cells 2 (*TREM2*), and **c.** their interaction. Different effects in the replication sample for **d.** *APOE*-ε4. **e.** Causal path model to estimate Tau_{Meta}. Solid lines indicate a direct effect of an upstream variable on a downstream pathology, dashed lines indicate an interactive effect between upstream variables (i.e. amyloid beta (Aβ), Tau_{EC}, or genetic variable) on meta temporal tau. Solid lines initiating from the black genetic variables box may indicate a direct or interactive effect of genetic variables on the outcome variable. Values in the parentheses indicate level of genetic variable (*APOE*-ε4 alleles (0,1,2) and *TREM2* risk variant carrier status (0,1)).

9. Regarding Figures 2-3: please add confidence intervals to the plots.

We now present in supplementary each marginal relationship with confidence intervals. We present each relationship as an independent panel for clarity and consistent presentation to ensure interpretability when many marginal effects are shown together (e.g., Figure 3 d).

Supplementary Figure 1 Genetic influences on Entorhinal Cortex (EC) tau. Lines show the estimated marginal levels of tau pathology for individuals with varying genetic profiles at different levels of A β , shaded portions represent the confidence intervals for the effect size across different levels of A β . Different effects in the discovery sample for **a.-c.** *APOE- ϵ 4*, **d.,e.** sex. Different effects in the replication sample for **f.-h.** *APOE- ϵ 4* and **i.,j.** sex. Values in the parentheses indicate level of genetic variable (*APOE- ϵ 4* alleles (0,1,2) and *TREM2* risk variant carrier status (0,1)).

Discovery Sample

Replication Sample

Supplementary Figure 2 Genetic influences on meta temporal tau. Lines show the estimated marginal levels of meta temporal (Tau_{Meta}) pathology for individuals with varying genetic profiles at different levels of entorhinal cortex tau (Tau_{EC}), shaded portions represent the confidence intervals for the effect size across different levels of Tau_{EC} . Different effects in the discovery sample for **a.,b.** *TREM2*, **c.-e.** *APOE- $\epsilon 4$* , **f.-k.** interaction of *TREM2* and *APOE- $\epsilon 4$* . Different effects in the replication sample for **i.-n.** *APOE- $\epsilon 4$* . Values in the parentheses indicate level of genetic variable (*APOE- $\epsilon 4$* alleles (0,1,2) and *TREM2* risk variant carrier status (0,1)).

References (as they appear in this document):

1. Colonna, M. The biology of TREM receptors. *Nat. Rev. Immunol.* 2023 239 **23**, 580–594 (2023).
2. Zhou, S. L. *et al.* TREM2 Variants and Neurodegenerative Diseases: A Systematic Review and Meta-Analysis. *J. Alzheimers. Dis.* **68**, 1171–1184 (2019).
3. Hardy, J. & Selkoe, D. J. The amyloid hypothesis of Alzheimer's disease: Progress and problems on the road to therapeutics. *Science (80-.).* **297**, 353–356 (2002).
4. Hampel, H. *et al.* The Amyloid- β Pathway in Alzheimer's Disease. *Mol. Psychiatry* 2021 2610 **26**, 5481–5503 (2021).
5. Bateman, R. J. *et al.* Clinical and Biomarker Changes in Dominantly Inherited Alzheimer's Disease. *N. Engl. J. Med.* **367**, 795–804 (2012).
6. Frisoni, G. B. *et al.* The probabilistic model of Alzheimer disease: the amyloid hypothesis revised. *Nat. Rev. Neurosci.* 2021 231 **23**, 53–66 (2021).
7. Jagust, W. J., Teunissen, C. E. & DeCarli, C. The complex pathway between amyloid β and cognition: implications for therapy. *Lancet. Neurol.* **22**, 847–857 (2023).
8. Belloy, M. E. *et al.* APOE Genotype and Alzheimer Disease Risk Across Age, Sex, and Population Ancestry. *JAMA Neurol.* **80**, 1284–1294 (2023).
9. Fortea, J. *et al.* APOE4 homozygosity represents a distinct genetic form of Alzheimer's disease. *Nat. Med.* 2024 305 **30**, 1284–1291 (2024).
10. Dang, C. *et al.* Relationship Between Amyloid- β Positivity and Progression to Mild Cognitive Impairment or Dementia over 8 Years in Cognitively Normal Older Adults. *J. Alzheimers. Dis.* **65**, 1313–1325 (2018).
11. Yamazaki, Y., Zhao, N., Caulfield, T. R., Liu, C. C. & Bu, G. Apolipoprotein E and Alzheimer disease: pathobiology and targeting strategies. *Nat. Rev. Neurol.* 2019 159 **15**, 501–518 (2019).
12. Deane, R. *et al.* apoE isoform-specific disruption of amyloid beta peptide clearance from mouse brain. *J. Clin. Invest.* **118**, 4002–4013 (2008).
13. Castellano, J. M. *et al.* Human apoE isoforms differentially regulate brain amyloid- β peptide clearance. *Sci. Transl. Med.* **3**, (2011).
14. Blumenfeld, J., Yip, O., Kim, M. J. & Huang, Y. Cell type-specific roles of APOE4 in Alzheimer disease. *Nat. Rev. Neurosci.* 2024 252 **25**, 91–110 (2024).
15. Jiang, Q. *et al.* ApoE Promotes the Proteolytic Degradation of A β . *Neuron* **58**, 681–693 (2008).
16. Brecht, W. J. *et al.* Neuron-specific apolipoprotein e4 proteolysis is associated with increased tau phosphorylation in brains of transgenic mice. *J. Neurosci.* **24**, 2527–2534 (2004).
17. Shi, Y. *et al.* ApoE4 markedly exacerbates tau-mediated neurodegeneration in a mouse model of tauopathy. *Nature* **549**, 523–527 (2017).
18. Ulrich, J. D., Ulland, T. K., Colonna, M. & Holtzman, D. M. Elucidating the Role of TREM2 in Alzheimer's Disease. *Neuron* **94**, 237–248 (2017).
19. Ulland, T. K. & Colonna, M. TREM2 - a key player in microglial biology and Alzheimer disease. *Nat. Rev. Neurol.* **14**, 667–675 (2018).
20. Hou, J., Chen, Y., Grajales-Reyes, G. & Colonna, M. TREM2 dependent and independent functions of microglia in Alzheimer's disease. *Mol. Neurodegener.* **17**, (2022).
21. Zhu, B. *et al.* Trem2 deletion enhances tau dispersion and pathology through microglia exosomes. *Mol. Neurodegener.* **17**, (2022).

22. Song, W. *et al.* Alzheimer's disease-associated TREM2 variants exhibit either decreased or increased ligand-dependent activation. *Alzheimer's Dement.* **13**, 381–387 (2017).
23. Sirkis, D. W. *et al.* Rare TREM2 variants associated with Alzheimer's disease display reduced cell surface expression. *Acta Neuropathol. Commun.* **4**, 98 (2016).
24. Carmona, S. *et al.* The role of TREM2 in Alzheimer's disease and other neurodegenerative disorders. *Lancet Neurol.* **17**, 721–730 (2018).
25. Beam, C. R. *et al.* Differences Between Women and Men in Incidence Rates of Dementia and Alzheimer's Disease. *J. Alzheimers. Dis.* **64**, 1077 (2018).
26. Filon, J. R. *et al.* Gender Differences in Alzheimer Disease: Brain Atrophy, Histopathology Burden, and Cognition. *J. Neuropathol. Exp. Neurol.* **75**, 748 (2016).
27. Liesinger, A. M. *et al.* Sex and age interact to determine clinicopathologic differences in Alzheimer's disease. *Acta Neuropathol.* **136**, 873 (2018).
28. Oveisgharan, S. *et al.* Sex differences in Alzheimer's disease and common neuropathologies of aging. *Acta Neuropathol.* **136**, 887 (2018).
29. Smith, R. *et al.* The accumulation rate of tau aggregates is higher in females and younger amyloid-positive subjects. *Brain* **143**, 3805–3815 (2020).
30. Buckley, R. F. *et al.* Sex Differences in the Association of Global Amyloid and Regional Tau Deposition Measured by Positron Emission Tomography in Clinically Normal Older Adults. *JAMA Neurol.* **76**, 542–551 (2019).
31. Buckley, R. F. *et al.* Sex Mediates Relationships Between Regional Tau Pathology and Cognitive Decline. *Ann. Neurol.* **88**, 921–932 (2020).
32. Edwards, L. *et al.* Multimodal neuroimaging of sex differences in cognitively impaired patients on the Alzheimer's continuum: greater tau-PET retention in females. *Neurobiol. Aging* **105**, 86–98 (2021).
33. Casaletto, K. B. *et al.* Sex-specific effects of microglial activation on Alzheimer's disease proteinopathy in older adults. *Brain* **145**, 3536 (2022).
34. CH, van D. *et al.* Lecanemab in Early Alzheimer's Disease. *N. Engl. J. Med.* **388**, 142–143 (2023).
35. Sims, J. R. *et al.* Donanemab in Early Symptomatic Alzheimer Disease: The TRAILBLAZER-ALZ 2 Randomized Clinical Trial. *JAMA* **330**, 512–527 (2023).
36. Kim, B. *et al.* TREM2 risk variants are associated with atypical Alzheimer's disease. *Acta Neuropathol.* **144**, 1085–1102 (2022).
37. Wang, C. *et al.* Gain of toxic apolipoprotein E4 effects in human iPSC-derived neurons is ameliorated by a small-molecule structure corrector. *Nat. Med.* **24**, 647–657 (2018).
38. Wadhvani, A. R., Affaneh, A., Van Gulden, S. & Kessler, J. A. Neuronal apolipoprotein E4 increases cell death and phosphorylated tau release in alzheimer disease. *Ann. Neurol.* **85**, 726–739 (2019).
39. Koutsodendris, N. *et al.* Neuronal APOE4 removal protects against tau-mediated gliosis, neurodegeneration and myelin deficits. *Nat. Aging* **2023 33 3**, 275–296 (2023).
40. Wang, C. *et al.* Selective removal of astrocytic APOE4 strongly protects against tau-mediated neurodegeneration and decreases synaptic phagocytosis by microglia. *Neuron* **109**, 1657-1674.e7 (2021).
41. Rao, A. *et al.* Microglia Depletion Reduces Human Neuronal APOE4-Driven Pathologies in a Chimeric Alzheimer's Disease Model. *bioRxiv* 2023.11.10.566510 (2023) doi:10.1101/2023.11.10.566510.
42. Haney, M. S. *et al.* APOE4/4 is linked to damaging lipid droplets in Alzheimer's

- disease microglia. *Nat.* 2024 6288006 **628**, 154–161 (2024).
43. Nuriel, T. *et al.* Neuronal hyperactivity due to loss of inhibitory tone in APOE4 mice lacking Alzheimer's disease-like pathology. *Nat. Commun.* 2017 81 **8**, 1–14 (2017).
 44. Giorgio, J., Adams, J. N., Maass, A., Jagust, W. J. & Breakspear, M. Amyloid induced hyperexcitability in default mode network drives medial temporal hyperactivity and early tau accumulation. *Neuron* **112**, 676–686.e4 (2024).
 45. Ferrari-Souza, J. P. *et al.* APOE ϵ 4 associates with microglial activation independently of A β plaques and tau tangles. *Sci. Adv.* **9**, (2023).
 46. Young, C. B. *et al.* APOE effects on regional tau in preclinical Alzheimer's disease. *Mol. Neurodegener.* **18**, 1–14 (2023).
 47. La Joie, R. *et al.* Association of APOE4 and Clinical Variability in Alzheimer Disease With the Pattern of Tau- and Amyloid-PET. *Neurology* **96**, e650–e661 (2021).
 48. Therriault, J. *et al.* Association of Apolipoprotein E ϵ 4 With Medial Temporal Tau Independent of Amyloid- β . *JAMA Neurol.* **77**, 470–479 (2020).
 49. Therriault, J. *et al.* APOE ϵ 4 potentiates the relationship between amyloid- β and tau pathologies. *Mol. Psychiatry* 2020 2610 **26**, 5977–5988 (2020).
 50. Ferrari-Souza, J. P. *et al.* APOE ϵ 4 potentiates amyloid β effects on longitudinal tau pathology. *Nat. Aging* 2023 310 **3**, 1210–1218 (2023).
 51. Stern, Y. *et al.* A framework for concepts of reserve and resilience in aging. *Neurobiol. Aging* **124**, 100–103 (2023).

Remaining Reviewer Comments:

Reviewer #2 (Remarks to the Author):

The manuscript, already good, has been improved based on the revisions. It would be nice for the authors to add comments from my previous point 10 into the discussion, as I think it is quite interesting and further supports the utility of the authors' model. But I do not think this is something that should hold up the publication process. The authors can make the decision on what they wish to do here.

We have now included comments regarding the reviewers previous point in the discussion. In particular the Discussion now writes pg 15

Using rare polymorphisms on the coding region of the *TREM2* gene, we find that trait differences in the function of TREM2 plays a role in the spread of tau from the EC into the neocortex (Figure 5 b.). Using causal path modelling we were able to infer at which stage genetic variation on the *TREM2* gene may become relevant in the AD cascade (i.e. before/after A β accumulation, or before/after EC tau accumulation). ...

Incorporating sTREM2 in a similar path modelling framework as a microglia response phenotype downstream to both genetic risk of *TREM2* and A β may provide additional state related changes in TREM2 further explaining the role of microglia in response to, and, in driving AD pathologies.